# On the relationship between disentanglement and multi-task learning

## Abstract

One of the main arguments behind studying disentangled representations is the assumption that they can be easily reused in different tasks. At the same time finding a joint, adaptable representation of data is one of the key challenges in the multi-task learning setting. In this paper, we take a closer look at the relationship between disentanglement and multi-task learning based on hard parameter sharing. We perform a thorough empirical study of the representations obtained by neural networks trained on automatically generated supervised tasks. Using a set of standard metrics we show that disentanglement appears naturally during the process of multi-task neural network training.

## 1 Introduction

Disentangled representations have recently become an important topic in the deep learning community (Eastwood & Williams, 2018; Locatello et al., 2019a; Ma et al., 2019; Sanchez et al., 2019; Do & Tran, 2020). The main assumption in this problem is that the data encountered in the real world is generated by few independent and explanatory factors of variation. It is commonly accepted that such representations are not only more interpretable and robust but also perform better in tasks related to transfer learning and one-shot learning (Bengio, 2013; Lake et al., 2017; Schölkopf et al., 2012; Locatello et al., 2019c).

Intuitively, a disentangled representation encompasses all the factors of variation and as such can be used for various tasks based on the same input space. On the other hand, non-disentangled representations, such as those learned by vanilla neural networks, might focus only on one or a few factors of variations that are relevant for the current task, while discarding the rest. Such a representation may fail when encountering different tasks that rely on distant aspects of variation which have not been captured.

Exploiting prevalent features and differences across tasks is also the paradigm of multi-task learning. In a standard formulation of a multi-task setting, a model is given one input and has to return predictions for multiple tasks at once. The neural network might be therefore implicitly regularized to capture more factors of variation than a network that learns only a single task. Based on this intuition, we hypothesize that disentanglement is likely to occur in the latent representations in this type of problem.

This paper aims to test this hypothesis empirically. We investigate whether the use of disentangled representations improves the performance of a multi-task neural network and whether disentanglement itself is achieved naturally during the training process in such a setting.

Our key contributions are:

- Construction of synthetic datasets that allow studying the relationship between multi-task and disentanglement learning.
- Study of the effect of multi-task learning with hard parameter sharing on the level of disentanglement obtained in the latent representation of the model.
- Analysis of the informativeness of the latent representation obtained in the single- and multi-task training.
- Inspection of the effect of disentangled representations on the performance of a multi-task model.

We verify our hypotheses by training multiple models in single- and multi-task settings and investigating the level of disentanglement achieved in their latent representations. In our experiments, we find that in a hard-parameter sharing scenario multi-task learning indeed seems to encourage disentanglement. However, it is inconclusive whether disentangled representations have a clear positive impact on the models performance, as the obtained by us results in this matter vary for different datasets.

## 2 RELATED WORK

### 2.1 DISENTANGLEMENT

Over the recent years, many methods that directly encourage disentanglement have been proposed. This includes algorithms based on variational and Wasserstein auto-encoders (Kim & Mnih, 2018; Higgins et al., 2017; Kumar et al., 2017; Brakel & Bengio, 2017; Spurek et al., 2020), flow networks (Dinh et al., 2014; Sorrenson et al., 2020) or generative adversarial networks (Chen et al., 2016). The main interest behind disentanglement learning lays in the assumption that such transformation unravels the semantically meaningful factors of variation present in the observations and thus it is desired in training deep learning models. In particular, disentanglement is believed to allow for informative compression of the data that results in a structural, interpretable representation, which is easily adaptable for new tasks (Bengio, 2013; Lake et al., 2017; Schmidhuber, 1992; Lipton, 2018).

Several of these properties have been experimentally proven in applications in many domains, including video processing tasks (Hsieh et al., 2018), recommendation systems (Ma et al., 2019) or abstract reasoning (Van Steenkiste et al., 2019; Steenbrugge et al., 2018). Moreover, recent research in reinforcement learning concludes that disentangling embeddings of skills allows for faster retraining and better generalization (Petangoda et al., 2019). Finally, disentanglement seems also to be positively correlated with fairness when sensitive variables are not observed (Locatello et al., 2019a). On the other hand, some empirical studies suggest that one should be cautious while interpreting the properties of disentangled representations. For instance, the latest studies in the unsupervised learning domain point that increased disentanglement does not lead to a decreased sample complexity in downstream tasks (Locatello et al., 2019b).

Another key challenge in studying disentangled representations is the fact that measuring the quality of the disentanglement is a nontrivial task (Do & Tran, 2020; Eastwood & Williams, 2018; Kim & Mnih, 2018), especially in a unsupervised setting (Locatello et al., 2019b). This motivates the research on practical advantages of disentanglement representations and their impact on the studied problem in possible future applications, which is the main focus of our work in the case of multi-task learning.

### 2.2 MULTI-TASK LEARNING

Multi-task learning aims at simultaneously solving multiple tasks by exploiting common information (Ruder, 2017). The approaches used predominantly to this problem are soft (Duong et al., 2015) and hard (Caruana, 1993) parameter sharing. In hard parameter sharing the weights of the model are divided into those shared by all tasks, and task-specific. In deep learning, this idea is typically implemented by sharing consecutive layers of the network, which are responsible for learning a joint data representation. In soft parameter sharing each task is given a set of separate parameters. The limitations are then imposed by information-sharing or regularizing the distance between the parameters by adding an applicable loss to the optimization objective.

Multi-task learning is widely used in the Deep Learning community, for instance in applications related to natural language processing (Liu et al., 2019), computer vision (Misra et al., 2016) or molecular property prediction modeled by graph neural networks (Capela et al., 2019). One may observe that the premises of multi-task and disentanglement learning are related to each other and thus it is interesting to investigate whether the joint data representation obtained in a multi-task problem exhibits some disentanglement-related properties.

## 3 METHODS

In this section, we describe the methods and datasets used for conducting the experiments.

### 3.1 DATASET CREATION

In order to investigate the relationship between multi-task learning and disentanglement, we require a dataset that fulfills two conditions:

1. It provides access to the true (disentangled) generative factors $z$ from which the observations $x$ are created.
2. It proposes multiple tasks for a supervised learner by providing labels $y_i$ which non-linearly depend on the true factors $z$.

The first condition is required in order to measure how well the learned representations approximate the true latent factors $z$. Access to the true factors allows for full control over the experimental settings and permits a fair comparison through the use of supervised disentanglement metrics. Note that even though unsupervised metrics have been proposed in the literature as well, they typically yield less reliable results, as we further discuss in section 3.3.

The second condition is needed to train a network on multiple nontrivial tasks to approximate the real-world setting of multi-task learning.

To our best knowledge, no nontrivial datasets exist that would abide by both those requirements. Most of the available disentanglement

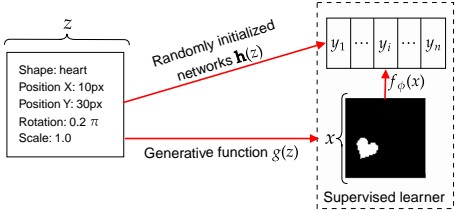

Figure 1: The setting of our experiments. Given a dataset of pairs $(x, z)$ of observations and their true generative factors, we generate a set of functions $\mathbf{h}(z)_i$ which are aimed to approximate real-world supervised tasks. Then, we train a neural network $f_\phi(x)$ in a multi-task regression setting on pairs $(x, \mathbf{h}(z))$. After the training, we investigate the hidden representations learned by $f_\phi$ and explore their relation to true factors $z$.

datasets, such as dSprites, Shapes3D, and MPI3D do fulfill the first condition, as they provide pairs $(x, z)$ of observations and their true generative factors. However, those datasets do not offer any type of challenging task on which our model could be trained. On the other hand, many datasets used for supervised multi-task learning fulfill the second condition by providing pairs $(x, y)$, but do not equip the researcher with the latent factors $z$ (ground truth), failing the first condition.

Thus, we aim to create our own datasets which fulfill both conditions by incorporating nontrivial tasks into standard disentanglement datasets. Since in multi-task approaches one often tries to solve tens of tasks at once, designing them by hand is infeasible and as such we decide to generate them automatically in a principled way. In particular, since supervised learning tasks might be formalized as finding a good approximation to an unknown function $h(x)$ given a set of points $(x, h(x))$, we generate random functions $h(z)$ which are then used to obtain targets for our dataset (see Figure 1).

We require $h(z)$ to be both nontrivial (i.e. non-linear and non-convex) and sufficiently smooth to approximate the nature of real-life tasks. In order to find a family of functions that fulfills those conditions, we take inspiration from the field of extreme learning, which finds that features obtained from randomly initialized neural networks are useful for training linear models on various real-world problems (Huang et al., 2011). As such, randomly initialized networks should be able to approximate these tasks up to a linear operation.

In particular, in order to generate the dataset, we define a neural network architecture $h(z, \theta)$. For this purpose, we used an MLP with four hidden layers with 300 units, tanh activations, and an output layer which returns a single number. Then we sample $n$ weight initializations of this network from the Gaussian distribution $\theta_i \sim \mathcal{N}(0, 1)$, where $i \in \{1, \dots, n\}$. Each of the networks $h(z, \theta_i)$ obtained by random initialization defines a single task in our approach. Thus, for a given dataset $\mathcal{D} = (x, z)$ containing observations and their true generative factors, we obtain a dataset for multi-task supervised learning by applying:

$$\tilde{\mathcal{D}} = \{(x, \mathbf{h}(z)) \mid (x, z) \in \mathcal{D}\} = \{(x, y)\},$$

where $\mathbf{h}(z)$ is a vector of stacked target values for each task, whose element $i$ is given by $\mathbf{h}(z)_i = h(z, \theta_i)$.

We use this data as a regression task, i.e. for a given neural network $f_\phi$ parameterized by $\phi$ the goal is to find:

$$\arg\min_\phi \sum_{(x,y)\in\tilde{\mathcal{D}}} \|f_\phi(x) - y\|_2^2.$$

We use this process to create multi-task supervised versions of dSprites, Shapes3D, and MPI3D, with 10 tasks for each dataset.

### 3.2 MODELS

#### 3.2.1 MULTI-TASK MODEL

We investigate the relation between disentanglement and multi-task learning based on a hard parameter sharing approach. In this setting, several consecutive hidden layers of the model are shared across all tasks in order to produce a joint data representation. This representation is then propagated to separate task-specific layers which are responsible for computing the final predictions.

In particular, we use a network consisting of a shared convolutional encoder and separate fully-connected heads for each of the tasks. The encoder learns the joint representation by transforming the inputs into a $d$-dimensional latent space. [1] The heads are implemented by 4-layer MLPs with ReLU activations, in order to match the capacity of the networks used for task generating functions $h_i(x)$. This overview of the model is illustrated in Figure 2.

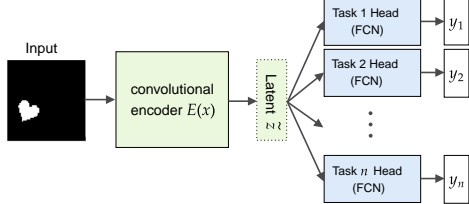

Figure 2: The model used for multi-task training. The convolutional encoder $E(x)$ transforms the input data $x$ to a latent representation $\tilde{z}$. The parameters of the encoder are shared across all tasks. Next, the produced representation is passed to the task-specific heads, which are implemented by fully-connected networks (FCN).

#### 3.2.2 AUTO-ENCODER MODEL

In the second part of our experiments we want to understand if disentangled representation provides some benefits for the multi-task problem. In order to produce disentangled representations, we decided to use three different representation-learning algorithms: a vanilla auto-encoder, the (beta)-variational auto-encoder (Kingma & Welling, 2013; Higgins et al., 2017) and FactorVAE (Kim & Mnih, 2018).

All these variants of the auto-encoder architecture encompass a similar framework. An auto-encoder imposes a bottleneck in the network which forces a compressed knowledge representation of the original input. In some variants of those models, we additionally try to constrain the latent variables to be highly informative and independent which further correlates to disentanglement, e.g. in models like $\beta$-VAE and FactorVAE. We use latent representations from these models to train task-specific heads and evaluate if disentanglement helped to decrease an error for that task.

The vanilla auto-encoder is also used in Section 4.2, where we add a decoder with transposed convolutions to pre-trained encoders from Section 4.1. This treatment is aimed to decode information for particular encoders in the most efficient way. As such, we find auto-encoders to be a useful tool for investigating disentanglement.

### 3.3 DISENTANGLEMENT METRICS

Measuring the qualitative and quantitative properties of the disentanglement representation discovered by the model is a nontrivial task. Due to the fact that the true generating factors of a given

---

[1]We provide the full model summary in **Appendix A**. The architecture of the encoder follows the one from (Abdi et al., 2019), which adopts the work of (Locatello et al., 2019b) for the `pytorch` package. We use the implementations from `https://github.com/amir-abdi/disentanglement-pytorch`.

dataset are usually unknown, one may assume that decomposition can be obtained only to some extend.

Commonly used unsupervised metrics are based on correlation coefficients which measure the intrinsic dependencies between the latent components. Such measures are widely used in the independent component analysis (Hyvarinen & Morioka, 2016; 2017; Hirayama et al., 2017; Brakel & Bengio, 2017; Spurek et al., 2020; Bedychaj et al., 2020). However, uncorrelatedness does not imply stochastical independence. Furthermore, metrics based on linear correlations may not be able to capture higher-order dependencies and are often ineffective in large dimensional or in over-determined spaces. All this makes the use of such unsupervised metrics questionable.

An alternative solution would be to use supervised metrics, which usually are more reliable (Locatello et al., 2019b). This is of course only possible after assuming access to the true generative factors. Such an assumption is rarely valid for real-world datasets, however, it is satisfied for synthetic datasets. Synthetic datasets present therefore a reasonable baseline for benchmarking disentanglement algorithms.

Frequently used metrics which use supervision are mutual information gap (MIG) (Chen et al., 2018), the FactorVAE metric (Kim & Mnih, 2018), Separated Attribute Predictability (SAP) score (Kumar et al., 2018) and disenanglement-completness-informativeness (DCI) (Eastwood & Williams, 2018). In order to comprehensively assess the level of disentanglement in our experiments, we have decided to use all of the above-mentioned metrics to validate our results. A more detailed description of those metrics is available in **Appendix B**.

## 4  RESULTS AND DISCUSSION

In this section, we describe the performed experiments and discuss the obtained results. For more details on the training regime and experimental setup please refer to **Appendix C**.

### 4.1  DOES HARD PARAMETER SHARING ENCOURAGE DISENTANGLEMENT?

One of the most common approaches to multi-task learning is hard parameter sharing. The key challenge in this method is to learn a joint representation of the data which is at the same time informative about the input and can be easily processed in more than one task. It is therefore tempting to verify whether disentanglement arises in those representations implicitly, as a consequence of hard parameter sharing.

In order to investigate this problem we build a simple multi-task model described in Section 3.2 and evaluate it on the three datasets discussed in Section 3.1: dSprites, Shapes3D, and MPI3D, each with 10 artificial tasks. After the training is complete, we calculate each of the disentanglement metrics described in Section 3.3 on the latent representation of the input data[2]. We compare the obtained results with the same metrics computed for an untrained (randomly initialized) network and for single-task models. In all the cases we use the same architecture and training regime. Note that in the single-model scenario we train a separate model for each of the 10 tasks, which is implemented by utilizing only one, dedicated head in the optimization process. We train all models three times, using a different random seed in the parameters initialization procedure. We report the mean results and standard deviations in Figure 3.

We observe that disentanglement metrics computed for the representations obtained in the multi-task setting are typically significantly better than the values obtained for single-task or random representations. Note that even the maximum mean result over all ten single-task models is in almost every case further than one standard deviation from the multitask mean. Moreover, this is true for all the tested datasets.

Let us also point out that instead of using separate heads for each of the tasks in the multi-task model one could simply use one head with the output dimension equal to the number of tasks and perform standard multivariate regression (with no parameter sharing). As presented in Figure 4, the latent representations emerging in such a scenario are less disentangled (in terms of the considered metrics)

---

[2]We use the implementations of Locatello et al. (2019b), which are available at `https://github.com/google-research/disentanglement_lib`

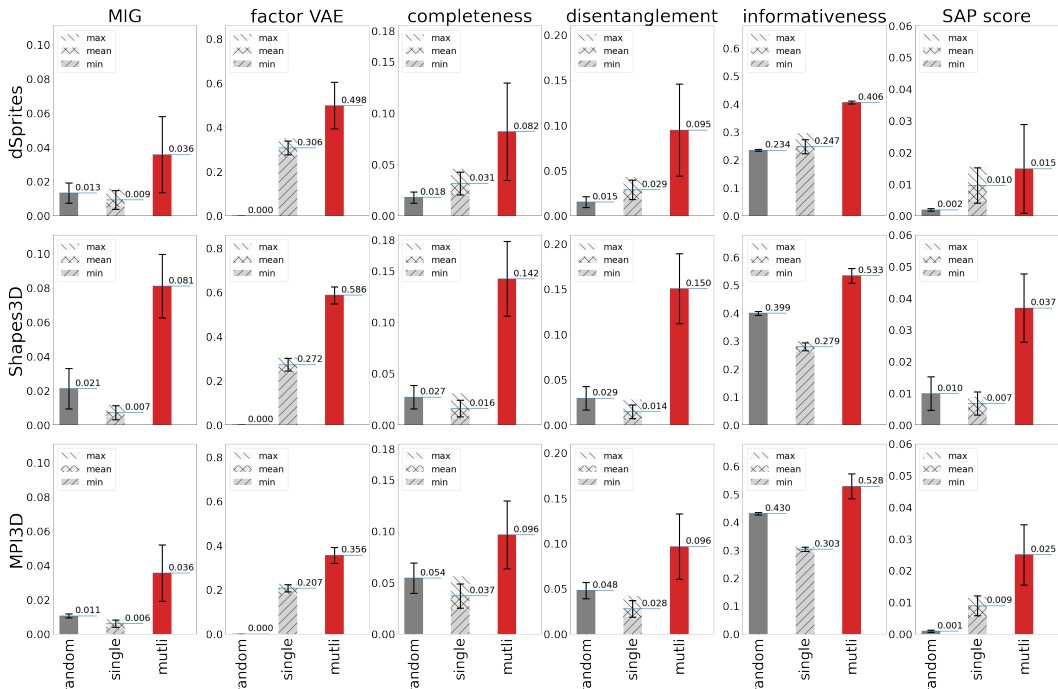

Figure 3: Different disentanglement metrics computed for random (untrained), single-task and multi-task models evaluated on the three datasets described in Section 3.1. The higher the value the better. For the single-task scenario, we report the mean over all task-specific models. Note that in almost every case the multi-task representations (red bars) outperform the random or single-task representations (dark-gray bars and light gray bars, respectively). Additionally, for single-task models, we report the maximal and minimal values over all tasks to show that the performance on multi-task does not rely on any single 'lucky' task. For tabulated results please refer to **Appendix E**.

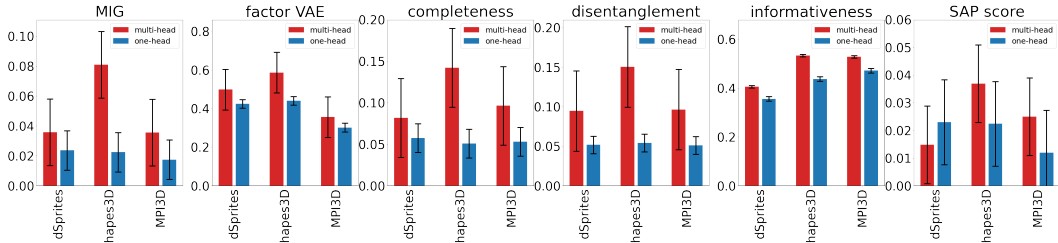

Figure 4: Different disentanglement metrics computed for the multi-task setting with one head shared between all tasks (one-head) and separate head for every task (multi-head), evaluated on the three datasets described in Section 3.1. The higher the value the better. One may observe that multi-head representations perform better than the ones obtained in the standard, one-head multivariate regression task. For tabulated results please refer to **Appendix E**.

than the representations obtained when utilizing hard parameter sharing. However, the achieved values are still better than in single-task models. This suggests that even though the increase in the metrics may be partially caused by simply training the network on higher-dimensional targets, the positive influence of hard parameter sharing cannot be ignored. This advocates in favor of the hypothesis that multi-task representations are indeed more disentangled than the ones arising in single-task learning.

## 4.2 What Are the Properties of the Learned Representations?

The previous section discussed the obtained representations by analyzing quantitative disentanglement metrics. Here, we provide more insights into the characteristics of latent encodings.

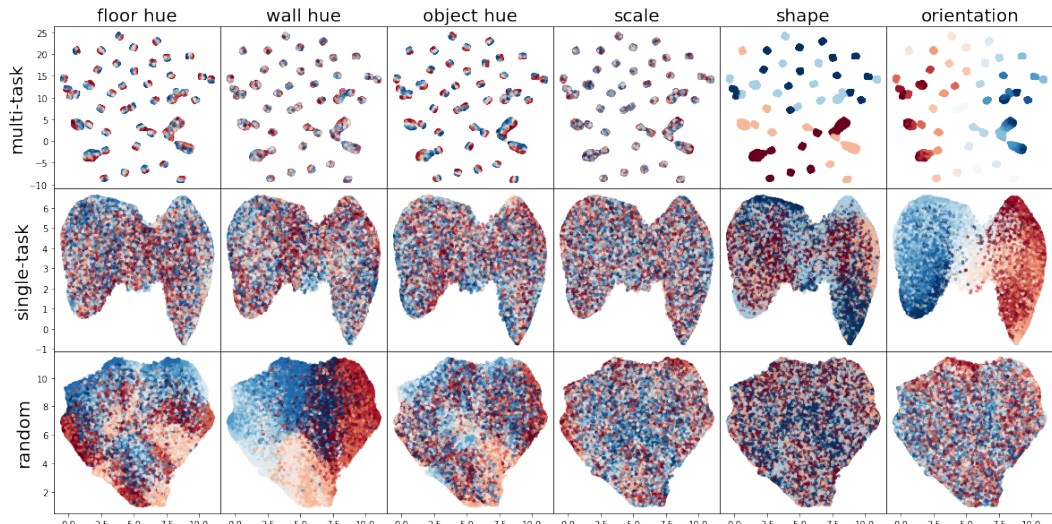

Figure 5: UMAP embeddings of the latent representations of the Shapes3D test dataset obtained for different models. Change of the color within one subplot presents the change in one particular ground truth component. The embeddings obtained by the multi-task model seem to be most semantically meaningful. See **Appendix D** for plots for other datasets.

### 4.2.1 UMAP EMBEDDINGS

In order to gain intuition behind the differences between the representations obtained in the previous experiment we compute a 2D-embedding of the latent encodings using the UMAP algorithm (McInnes et al., 2018). The results are presented in Figure 5.

The embeddings obtained for the multi-task representations are much more semantically meaningful, with easily distinguishable separate clusters. Moreover, the position and internal structure of the clusters correspond to different values of the true factors. This cannot be observed for the untrained or single-task representations, suggesting that the multi-task representations are indeed more successful in encompassing the information about the real values of the generative sources of the data.

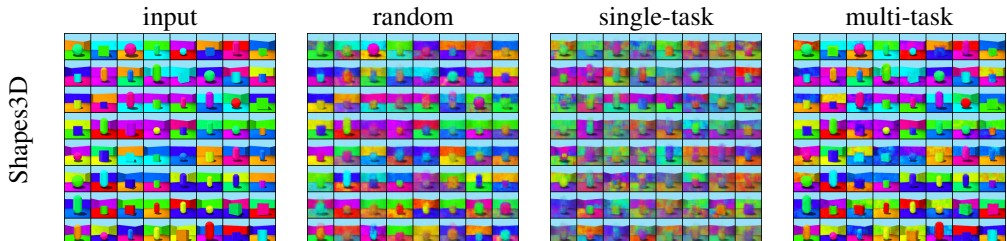

Figure 6: Reconstructions obtained by the decoders trained on random, single-task, and multi-task encoding. For reference, we provide the original input images in the first row. The quality of the reconstruction for the random and single-task representation is very poor. Contrary, the multi-task encoder provided a latent space that can be successfully decoded into images that closely resemble the corresponding examples from the input. Thus, we conclude that the multi-task representations are more informative about the data and provide better compression. See **Appendix D.2** for reconstructions for other datasets.

### 4.2.2 LATENT SPACE TRAVERSAL

Providing qualitative results of the retrieved factors is a common practice in disentanglement learning (Locatello et al., 2019c; Kumar et al., 2017; Sanchez et al., 2019; Sorrenson et al., 2020; Locatello et al., 2019b). In particular, visual presentation of the interpolations over the latent space allows assessing — from a human perspective — the informativeness and decomposition of the ob-

tained representations. Note that such analysis is possible only after adding and training a suitable decoder network, which maps the retrieved factors back to the image space.

In our setting, the decoder mirrors the architecture of the encoder (the convolutions are replaced by transposed convolutions of the same size — see **Appendix A**). Given the latent representations as an input, the decoder optimizes the reconstruction error (as measured by MSE) between its outputs and the original images. We train three separate decoders corresponding to the different encoders from the previous section — a randomly initialized encoder, an encoder produced by one of the single-task models, and a multi-task encoder.

First, let us discuss the reconstruction quality achieved by each of the tested decoders. Results of this experiment are presented in Figure 6[3]. Reconstructions produced for the multi-task encodings are clearly superior to the ones obtained for the single-task encodings. In the first case, the resulting images are sharp and contain almost no noise. In contrast, the single task reconstructions are blurry and similar to the ones produced for the randomly initialized encoder. We would like to emphasise that all the decoders used the same architecture and that during their optimization the parameters of the corresponding encoders were kept fixed. Therefore the quality of the reconstruction is an important property of a latent representation, as it allows us to assess the compression capacity of the representation. From this perspective, the compression obtained in the multi-task scenario is much more informative about the input than in the single-task scenario.

Another approach to the visualisation of the latent variables is to perform interpolations (traversals) in the latent space. We start by selecting a random sample from the dataset and compute its encoding $\tilde{z} \in \mathbb{R}^d$. By modifying one of the components of vector $\tilde{z}$ from $-1$ to $1$ with $0.1$ step and leaving the $d-1$ unchanged, we produce a traversal along that particular factor. We repeat this procedure for all the factors in order to capture their impact on the decoded example. Results of such traverses for the dSprites dataset are shown in Figure 7.

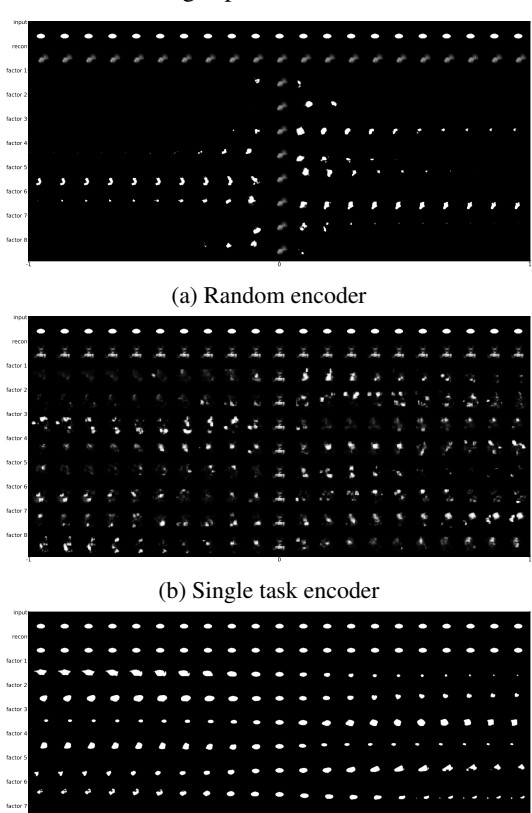

(a) Random encoder

(b) Single task encoder

(c) Multi-task encoder

Figure 7: Traverses over latent variable produced for a given architecture. The same example was used in all three traverses. The second row of each image shows how the decoder reconstructed this example in a particular setting. The rest of the factors come from latent space generated from each encoder. Visualization of components from the multi-task encoder are sharp and distinguish the generating factors distinctly. The same cannot be said about the latent factors in single-task and random encoders, which are blurry and disconnected from any interpretable ground truth factors. Please refer to **Appendix D** for the results of the traversals over other datasets.

Note that since the models were not trained directly for disentanglement but only to solve a supervision task, it is not surprising that the representations are not as clearly factorized as in specialized methods such as FactorVAE. However, for the multi-task model, certain latent dimensions still appear to be disentangled and one can easily spot the difference in quality between the single and multi-task representations. In the multi-task traversals, we can notice components that are responsible for the position and scale of a given figure (in Figure 7c, consider the 5th and 7th factors, respectively). In contrast, the results for single task representations demonstrate that even a slight change in any of the single latent dimensions leads to a degradation of the reconstructed examples.

---

[3]Numerical values for reconstruction errors are presented in **Appendix D.2.**

As expected, this effect is even more evident for the random (untrained) representations, where the corruption over latent factor is even more prevalent than in the case of a single-task traversal.

### 4.3 DOES DISENTANGLEMENT HELP IN TRAINING MULTI-TASK MODELS?

In the previous sections, we studied whether multi-task learning encourages disentanglement. Here we consider an inverse problem by asking whether using disentangled representation helps in multi-task learning. To investigate this issue, we train an auto-encoder-based model devised specifically to produce disentangled latent representations without access to the true latent factors. Next, we freeze its parameters and use the encoder function to transform the inputs. The obtained latent encodings are then passed directly to the heads of a multi-task network which minimizes the average regression loss given the target values of the artificial tasks.

We consider three different auto-encoder-based algorithms described in Section 3.2.2: a vanilla auto-encoder (AE), a variational auto-encoder (VAE), and the FactorVAE. The vanilla auto-encoder does not directly enforce latent disentanglement during the training. In the VAE model, the prior normal distribution with identity covariance matrix implies some disentanglement. Finally, FactorVAE introduces a new module to the VAE architecture that explicitly induces informative decomposition. Therefore, the representations obtained for each subsequent model should be also naturally ordered by the level of the achieved disentanglement. For the exact values of the calculated metrics please refer to **Appendix F**. In addition, we also study a scenario in which we explicitly provide the true source factors. We trained all regression models three times, using a different random seed in the parameters initialization procedure.

Table 1 summarizes the performance of the multi-task model trained on the representations obtained for the above-discussed methods. Although the representations obtained from FactorVAE are better (see, for instance, MIG or DCI measures in **Appendix F**) than those from VAE and AE, the encodings produced by the vanilla AE are the best among the tested, exceeding the others on Shapes3D and MPI3D and

Table 1: RMSE of multi-task networks trained on latent representations obtained by different auto-encoder-based methods. For comparison, we added the model trained on ground truth factors. The best results are bolded, and best out of auto-encoder architectures underlined.

| Dataset | dSprites | Shapes3D | MPI3D |
|---|---|---|---|
| Ground Truth | $150.235 \pm 3.754$ | $\mathbf{72.979 \pm 0.193}$ | $\mathbf{108.568 \pm 0.285}$ |
| AE | $80.062 \pm 0.341$ | $\underline{114.939 \pm 0.160}$ | $\underline{150.190 \pm 0.097}$ |
| VAE | $\underline{\mathbf{63.260 \pm 0.260}}$ | $132.072 \pm 0.169$ | $194.865 \pm 15.61$ |
| FactorVAE | $91.937 \pm 0.199$ | $118.396 \pm 0.423$ | $151.646 \pm 0.336$ |

being second on dSprites. Note that these results coincide with observations presented in the literature. For example, (Locatello et al., 2019b) compared different models that enforce disentanglement during the training and showed that even a high value of that property within the factors do not constitute a better model performance. However, in two out of three datasets, the use of the ground true factors seems to significantly improve the obtained results. This may suggest that the representations produced by the considered disentanglement methods are not fully factorized. It is therefore inconclusive whether the discrepancy between the obtained results is due to the shortcomings of the used methods or a manifestation of the impracticality of disentanglement.

## 5 CONCLUSIONS

In this paper, we studied the relationship between multi-task and disentanglement representation learning. A fair evaluation of our hypothesis is impossible on real-world datasets, without provided ground truth factors. To evaluate our results we had to introduce synthetic datasets that contain all necessary properties to be seen as a benchmark in this field. Next, we studied the effects of multi-task learning with hard parameter sharing on representation learning. We found that nontrivial disentanglement appears in the representations learned in a multi-task setting. Obtained factors have intuitive interpretations and correspond to the actual ground truth components. Finally, we inverted the question and investigated the hypothesis that disentangled representation is needed for multi-task learning, the results however are not conclusive. We found out that multi-task models benefit from disentanglement only on specific datasets. However, we cannot name an indicator of when this unambiguously applies.

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

## A    SUMMARY OF THE ARCHITECTURE OF THE MULTI-TASK MODEL

The architecture of the convolutional encoder $E(x)$ is provided in Table 2, together with the architecture of the corresponding decoder, which was used in experiments in Section 4.2. For the fully-connected heads, we used the same architecture as the one utilized during dataset creation, which is presented in Table 3.

Table 2: The architecture of auto-encoder-based methods. Non-linearity in all layers is given by ReLU function.

| Encoder | | | | Decoder | | | |
|---|---|---|---|---|---|---|---|
| Type | Kernel | Stride | Outputs | Type | Kernel | Stride | Outputs |
| Conv 2d | 4 | 2 | 32 | Conv 2d | 1 | 2 | 256 |
| Conv 2d | 4 | 2 | 32 | Conv Transpose 2d | 4 | 2 | 256 |
| Conv 2d | 4 | 2 | 64 | Conv Transpose 2d | 4 | 2 | 128 |
| Conv 2d | 4 | 2 | 128 | Conv Transpose 2d | 4 | 2 | 128 |
| Conv 2d | 4 | 2 | 256 | Conv Transpose 2d | 4 | 2 | 64 |
| Conv 2d | 4 | 2 | 256 | Conv Transpose 2d | 4 | 2 | 64 |
| Dense | | | output_dim | Conv Transpose 2d | 3 | 1 | num_channels |

Table 3: The architecture of a single fully-connected head in the single- and multi-task neural network. We apply non-linearity (given by the ReLU function) after all layers except the last one.

| Type | Output shape |
|---|---|
| Dense | 300 |
| Dense | 300 |
| Dense | 300 |
| Dense | 10 |

## B    DISENTANGLEMENT METRICS

In our experiments, we decided to use four measures of disentanglement to comprehensively validate our results. For the convenience of the reader, in this part of the appendix, we shortly describe the used measures (for wider context we encourage the reader to refer to the original papers).

### B.1    MUTUAL INFORMATION GAP (MIG)

MIG computes the mutual information between each of the ground truth components $z_i$ and the disentangled factor $\tilde{z}_j$. The mutual information between $z_i$ and $\tilde{z}_j$ is denoted by $I(z_i, \tilde{z}_j)$. Next, the latent dimension with maximum mutual information score is identified for each of the retrieved factor (denoted by $I(z_i, \tilde{z}_{\max_1})$), along with the second-best result of the same score (denoted by $I(z_i, \tilde{z}_{\max_2})$). The difference between those values gives a gap, which finally is normalized with respect to the total mutual information associated with the studied factor:

$$\text{MIG} = \frac{I(z_i, \tilde{z}_{\max_1}) - I(z_i, \tilde{z}_{\max_2})}{\sum_{j=1}^{m} I(z_i, \tilde{z}_j)}.$$

Where $m$ is the dimension of ground truth components space. To report one score we average the MIG scores of all factors.

### B.2    FACTORVAE METRIC

We start by normalizing retrieved factors by their respective standard deviation computed over the dataset. For a subset of the dataset, a ground truth component is then randomly selected and fixed at a random value. Variance is then computed over normalized retrieved factors in this subset. Next,

the lowest variance factor — the one that should mostly resemble the fixed ground truth component — is associated with that ground truth component.

This procedure with selecting the subsets and fixing one of its ground truth components is then repeated multiple times (in our experiments 10000 times). As a result, the associations between disentangled factor and ground truth component are used as inputs in a majority vote classifier. FactorVAE metric is the mean accuracy of the classifier.

### B.3 Separated Attribute Predictability (SAP)

SAP attributes a score $S_{ij}$ to all pairs of ground truth components $z_i$ and disentangled factors $\tilde{z}_j$. For continuous components, linear regression predicts the disentangled factors, and $S_{ij}$ is the coefficient of determination ($R^2$) of the regression. In the case of categorical features, SAP fits a decision tree on ground truth components and reports the balanced classification accuracy. The final SAP score is achieved by computing the difference between the two highest $S_{ij}$ values for all factors:

$$SAP = \frac{1}{n} \sum_{i=1}^{n} S_{i\max_1} - S_{i\max_2},$$

where $n$ is the dimension of ground truth components space, $S_{i\max_1}$ is the highest score for component $z_i$ and $S_{i\max_2}$ is the second highest score for the same component.

### B.4 Disentanglement, Completeness, and Informativeness (DCI)

Unlike previous measures, DCI is a complete framework that allows verifying several properties of the achieved representation. Disentanglement and completeness are estimated by inspecting the regressor's parameters to derive predictive importance weights $R_{ij}$ for each pair $(z_i, \tilde{z}_j)$ of ground truth $z_i$ and retrieved $\tilde{z}_j$ components.

The completeness for ground truth component $z_i$ is given by

$$C_i = 1 + \sum_{j=1}^{m} p_{ij} \log_n p_{ij},$$

where $m$ stands for ground truth dimension and $p_{ij}$ is the probability that disentangled factor $\tilde{z}_j$ is important to predict $z_i$. These probabilities are obtained by dividing each importance weight by the sum of all importance weights related to a given component:

$$p_{ij} = \frac{R_{ij}}{\sum_{k=1}^{m} R_{ik}}.$$

The final compactness score is an average of compactness scores over all components.

Disentanglement for retrieved factor $\tilde{z}_j$ is given by

$$D_j = 1 + \sum_{i=1}^{d} p_{ij} \log_d p_{ij}$$

where $d$ is the dimension of the latent space and $p_{ij}$ is the probability that the latent factor $\tilde{z}_j$ is important to predict only the component $z_i$. Analogously to completeness, those probabilities are normalized with respect to potentially disentangled factors:

$$p_{ij} = \frac{R_{ij}}{\sum_{k=1}^{d} R_{kj}}.$$

The final disentanglement score is a weighted average of the individual disentanglement scores:

$$D = \sum_{j=1}^{n} \mu_j D_j, \text{ where } \mu_j = \frac{\sum_{i=1}^{d} R_{ij}}{\sum_{k=1}^{n} \sum_{i=1}^{d} R_{ik}}.$$

If a disentangled variable $\tilde{z}_i$ is irrelevant for predicting $z_j$, then its $\mu_i$ (and thus contribution to the overall disentanglement) will be near zero.

Finally, the prediction error of the regressor measures the informativeness of the representation. Normalized inputs and outputs allow to compute the estimation error for a completely random mapping and use it to normalize the score between $0$ and $1$.

## C  TRAINING REGIME AND EXPERIMENTAL SETUP

### C.1  THE MULTI-TASK MODEL — EXPERIMENT 4.1

We train the multi-task model to minimize the sum of the task errors. The training is performed for 200 epochs with learning rate 0.001 and batch size 256, by using the AdaM optimizer (Kingma & Ba, 2014) with $\beta_1 = 0.9$ and $\beta_2 = 0.999$. We repeat this procedure three times, changing the random seed initialization, and report the mean and average values of the disentanglement metrics.

### C.2  LATENT VISUALISATIONS — EXPERIMENT 4.2

The encoder architecture was taken from the experiments in Section 4.3. The multi-task model for each experiment was randomly selected from one of the seeds from the 10 tasks setting. Additionally, one of the single-task encoders was selected out of the trained ones for the same seed. The random encoder was initialized by the default initialization used by the `pytorch` library.

The decoder architecture was optimized by minimizing the mean square error between the decoded and input image. The training was performed over 500 epochs. We used mini-batches of 64 images and gradually reduced learning rate starting from 0.0002, with a reduction of 50% every 100 epochs.

### C.3  CLASSIFICATION BASED ON LATENT FACTORS — EXPERIMENT 4.3

We used the same auto-encoder and multi-task architectures like the one used in previous experiments (and defined in Section A), however with non-linearity given by tanh function. We trained all auto-encoders for 100 epochs, using batch size 64, learning rate 0.0001, Adam optimizer (Kingma & Ba, 2014) and latent dimension equal to 8. Other hyperparameters settings were adapted from (Abdi et al., 2019). Multi-task networks were trained for 30 epochs, using batch size 64, learning rate 0.0001, and adam optimizer. In order to average the scores over different runs, we repeated the multi-task network training 3 times.

## D  VISUALISATIONS OF DECODED REPRESENTATIONS

### D.1  UMAP EMBEDDINGS

In order to visualize the latent representations obtained for the random (untrained), single-task, and multi-task models we embed them into a two-dimensional space by using the UMAP algorithm. The results are shown in Figure 8. It may be observed that the embeddings obtained for the multi-task representations are much more semantically meaningful. This is especially evident for the dSprites and Shapes3D datasets. The MPI3D dataset is a significantly more difficult problem, and although the multi-task embeddings seem to be correlated to some of the true factors, the difference is not as visible in this case.

### D.2  RECONSTRUCTIONS

As described in Section 4.2, we trained decoders over various latent spaces produced by the encoders in the experiment from Section 4.1. We provide the numerical values of the reconstruction error in Table 4 and qualitative images of the reconstructed examples in Figure 9. It may be observed that the latent representations produced by random and single task encoders do not allow the decoder to successfully restore the input examples. Moreover, the decoder trained on single-task latent is even worse (in the case of reconstruction) than the random one.

Table 4: Test reconstruction error between the decoded images and the original input images.

|  | random | single-task | multi-task |
|---|---|---|---|
| dSprites | 308.04 | 326.30 | 35.97 |
| Shapes3D | 0.044 | 0.082 | 0.008 |
| MPI3D | 0.0021 | 0.0061 | 0.0009 |

Table 5: The exact values of the metrics computed in the experiment from Section 4.1.

(a) MIG

| model | dSprites | Sbapes3D | MPI3D |
|---|---|---|---|
| random | $0.01 \pm 0.01$ | $0.02 \pm 0.01$ | $0.01 \pm 0.00$ |
| single-mean | $0.01 \pm 0.01$ | $0.01 \pm 0.00$ | $0.01 \pm 0.00$ |
| single-max | $0.02 \pm 0.00$ | $0.01 \pm 0.00$ | $0.01 \pm 0.00$ |
| single-min | $0.01 \pm 0.00$ | $0.00 \pm 0.00$ | $0.00 \pm 0.00$ |
| multi-head | $\mathbf{0.04 \pm 0.02}$ | $\mathbf{0.08 \pm 0.02}$ | $\mathbf{0.04 \pm 0.02}$ |
| one-head | $0.02 \pm 0.01$ | $0.02 \pm 0.01$ | $0.02 \pm 0.00$ |

(b) Factor VAE metric

| model | dSprites | Sbapes3D | MPI3D |
|---|---|---|---|
| random | $0.00 \pm 0.00$ | $0.00 \pm 0.00$ | $0.00 \pm 0.00$ |
| single-mean | $0.31 \pm 0.03$ | $0.27 \pm 0.03$ | $0.21 \pm 0.02$ |
| single-max | $0.35 \pm 0.04$ | $0.31 \pm 0.01$ | $0.23 \pm 0.04$ |
| single-min | $0.26 \pm 0.01$ | $0.23 \pm 0.01$ | $0.18 \pm 0.01$ |
| multi-head | $\mathbf{0.50 \pm 0.11}$ | $\mathbf{0.59 \pm 0.04}$ | $\mathbf{0.36 \pm 0.04}$ |
| one-head | $0.42 \pm 0.02$ | $0.44 \pm 0.06$ | $0.30 \pm 0.04$ |

(c) completeness (DCI)

| model | dSprites | Sbapes3D | MPI3D |
|---|---|---|---|
| random | $0.02 \pm 0.01$ | $0.03 \pm 0.01$ | $0.05 \pm 0.01$ |
| single-mean | $0.03 \pm 0.01$ | $0.02 \pm 0.01$ | $0.04 \pm 0.01$ |
| single-max | $0.05 \pm 0.02$ | $0.03 \pm 0.01$ | $0.06 \pm 0.01$ |
| single-min | $0.02 \pm 0.00$ | $0.01 \pm 0.00$ | $0.02 \pm 0.00$ |
| multi-head | $\mathbf{0.08 \pm 0.05}$ | $\mathbf{0.14 \pm 0.04}$ | $\mathbf{0.10 \pm 0.03}$ |
| one-head | $0.06 \pm 0.02$ | $0.05 \pm 0.02$ | $0.05 \pm 0.01$ |

(d) disentanglement (DCI)

| model | dSprites | Sbapes3D | MPI3D |
|---|---|---|---|
| random | $0.02 \pm 0.01$ | $0.03 \pm 0.01$ | $0.05 \pm 0.01$ |
| single-mean | $0.03 \pm 0.01$ | $0.01 \pm 0.01$ | $0.03 \pm 0.01$ |
| single-max | $0.04 \pm 0.01$ | $0.03 \pm 0.01$ | $0.04 \pm 0.01$ |
| single-min | $0.02 \pm 0.01$ | $0.01 \pm 0.00$ | $0.02 \pm 0.00$ |
| multi-head | $\mathbf{0.09 \pm 0.05}$ | $\mathbf{0.15 \pm 0.04}$ | $\mathbf{0.10 \pm 0.04}$ |
| one-head | $0.05 \pm 0.01$ | $0.05 \pm 0.03$ | $0.05 \pm 0.01$ |

(e) informativeness (DCI)

| model | dSprites | Sbapes3D | MPI3D |
|---|---|---|---|
| random | $0.23 \pm 0.00$ | $0.40 \pm 0.01$ | $0.43 \pm 0.00$ |
| single-mean | $0.25 \pm 0.03$ | $0.28 \pm 0.01$ | $0.30 \pm 0.01$ |
| single-max | $0.30 \pm 0.02$ | $0.30 \pm 0.01$ | $0.31 \pm 0.01$ |
| single-min | $0.23 \pm 0.01$ | $0.26 \pm 0.01$ | $0.29 \pm 0.00$ |
| multi-head | $\mathbf{0.41 \pm 0.01}$ | $\mathbf{0.53 \pm 0.03}$ | $\mathbf{0.53 \pm 0.04}$ |
| one-head | $0.36 \pm 0.01$ | $0.44 \pm 0.03$ | $0.47 \pm 0.04$ |

(f) SAP score

| model | dSprites | Sbapes3D | MPI3D |
|---|---|---|---|
| random | $0.00 \pm 0.00$ | $0.01 \pm 0.01$ | $0.00 \pm 0.00$ |
| single-mean | $0.01 \pm 0.01$ | $0.01 \pm 0.00$ | $0.01 \pm 0.00$ |
| single-max | $\mathbf{0.02 \pm 0.00}$ | $0.01 \pm 0.00$ | $0.01 \pm 0.00$ |
| single-min | $0.00 \pm 0.00$ | $0.00 \pm 0.00$ | $0.01 \pm 0.00$ |
| multi-head | $0.01 \pm 0.01$ | $\mathbf{0.04 \pm 0.01}$ | $\mathbf{0.02 \pm 0.01}$ |
| one-head | $0.02 \pm 0.02$ | $0.02 \pm 0.01$ | $0.01 \pm 0.01$ |

## D.3 TRAVERSALS IN LATENT SPACE

In parallel to the study of the quality of the reconstructions, we have also explored the traversals in latent spaces. Given a latent representation $\tilde{z}$ of an arbitrary image $x$ we compute the traverse along each one of the components of $\tilde{z}$, as described in Section 4.2. This traversal represents how the image changes if only one component is slightly modified. This procedure provides a visually qualitative way of assessing the level of disentangled in the obtained representations.

In order to complement the discussion conducted in Section 4.2 we present here also the traversals for the Shapes3D and MPI3D datasets (in Figures 11 and 12, respectively). One may observe that the results align with the quantitative studies of disentangled metrics from Figure 3 — where we showed that the most disentangled representation is obtained in the multi-task scenario. Note that the most informative changes of a particular feature for a given object may be observed in multi-task traversals. One may spot that object factors — although not totally disentangled — change independently from each other.

The same is not true for single-task traversals. In the example from Shapes3D dataset (Figure 11), we observe that the single-task traversals capture only the color change of the background wall. It is also not surprising that the least informative traversal comes from the randomly initialized encoder.

## E   DISENTANGLEMENT AND HARD PARAMETER SHARING

In Section 4.1 we discuss the influence of hard parameter sharing on disentanglement learning. Here we present the computed metrics for all models (including regression) in a tabulated manner in Table 5. In addition, we also present the average MSE loss on the test dataset in Figure 13.

## F   DISENTANGLED REPRESENTATION AS BASES FOR MULTI-TASK TRAINING

In Section 4.3 we took the opportunity to discuss how disentanglement influences multi-task training. In this section, we present numerical results of all computed disentanglement metrics across trained encoders. It is not surprising that FactorVAE representations are most disentangled in the

Table 6: Numerical results of disentanglement metrics for latent on which multi-task training was performed.

(a) MIG

| Dataset | dSprites | Shapes3D | MPI3D |
|---|---|---|---|
| AE | 0.028 | 0.028 | 0.023 |
| VAE | 0.117 | 0.041 | 0.011 |
| FactorVAE | **0.272** | **0.251** | **0.040** |

(b) SAP score

| Dataset | dSprites | Shapes3D | MPI3D |
|---|---|---|---|
| AE | 0.006 | **0.020** | 0.009 |
| VAE | 0.032 | **0.020** | **0.017** |
| FactorVAE | **0.068** | 0.020 | 0.011 |

(c) Factor VAE metric

| Dataset | dSprites | Shapes3D | MPI3D |
|---|---|---|---|
| AE | 0.566 | 0.565 | 0.297 |
| VAE | **0.710** | **0.564** | **0.323** |
| FactorVAE | 0.622 | 0.690 | 0.310 |

(d) informativeness (DCI)

| Dataset | dSprites | Shapes3D | MPI3D |
|---|---|---|---|
| AE | 0.395 | 0.493 | 0.473 |
| VAE | 0.579 | 0.533 | **0.484** |
| FactorVAE | **0.664** | **0.610** | 0.482 |

(e) disentanglement (DCI)

| Dataset | dSprites | Shapes3D | MPI3D |
|---|---|---|---|
| AE | 0.052 | 0.081 | 0.070 |
| VAE | 0.257 | 0.124 | **0.119** |
| FactorVAE | **0.356** | **0.342** | 0.079 |

(f) completeness (DCI)

| Dataset | dSprites | Shapes3D | MPI3D |
|---|---|---|---|
| AE | 0.046 | 0.078 | 0.078 |
| VAE | 0.271 | 0.120 | **0.128** |
| FactorVAE | **0.407** | **0.331** | 0.091 |

predominant number of cases. What can be read as a surprise is that FactorVAE representations are never the best in terms of the root mean square error metric of the model that was trained on them.

## G  INCREASING THE NUMBER OF TASKS

Apart from the tested in the main paper scenario with 10 tasks, we also conducted experiments with varying number of tasks $n$ from the list of $[5, 10, 20, 30, 40, 50]$. The results are presented in Figure 14. It is impossible to draw any clear conclusions from this results, as the results vary a lot. It may be observed that in some cases increasing the number of tasks up to 30 leads to higher values of selected metrics, but at the same time having a negative impact on the others (for instance, Shapes3D and Factor_VAE versus DCI disentanglement or DCI completeness). These discrepancies are also not consistent between datasets (consider the top row for the dSprites dataset versus the Shapes3D).

## H  VARYING THE NUMBER OF USED GENERATING FACTORS

Apart from the presented in Section 4.3 approach which uses all of the factors to generate a task, we also considered a scenario in which a random subset of the factors is sampled for each task, and a scenario in which the tasks are generated from disjoint subsets (every odd task depends only on the first half of the factors and every even task on the other half). We compare these approaches in Figure 15. The computed disentanglement measures vary and the precise subset of incorporated factors in the task generating procedure does not have any conclusive impact on the final quality of the learned representation.

## I  NUMBER OF RETRIEVED COMPONENTS

In addition to the results presented in Section 4.3 we also compute the number of retrieved components and the mean correlation values between the retrieved components and the ground truth factors in Table 7. The results are computed for the representations obtained on the test splits for each datasets used in the UMAP embedding experiment in Section 4.3. To get the number of retrieved components for each of the component of the representation we compute the spearman correlation with each of the ground true factor and choose the one for which the correlation is the largest and statistically significant. We next return the number of unique components matched in this way.

| dataset | factors retrived | mean corr | std corr | min corr | max corr |
|---|---|---|---|---|---|
| dsprites_multi | 4 | 0.385113 | 0.166741 | 0.079401 | 0.616765 |
| dsprites_single | 4 | 0.165147 | 0.057721 | 0.085172 | 0.270090 |
| shapes3d_multi | 5 | 0.518252 | 0.307997 | 0.041420 | 0.903151 |
| shapes3d_single | 4 | 0.311432 | 0.287550 | 0.000000 | 0.793141 |
| mpi3d_multi | 6 | 0.317056 | 0.142792 | 0.111870 | 0.585552 |
| mpi3d_single | 5 | 0.202390 | 0.111612 | 0.061758 | 0.363220 |

Table 7: The number of factors retrived by each method (`mulit` for multi-task models and `single` for a single task models) and the average/std/min and max correlation of the retrieved components with the ground truth factors.

## J  PERFORMANCE ON SINGLE TASKS

In this section we provide the test losses on all tasks in Tables 8, 9, and 10 for the dSprites, Shapes3D, and MPI3D datasets, respectively.

| task | loss0 | loss1 | loss2 | loss3 | loss4 | loss5 | loss6 | loss7 | loss8 | loss9 | total_loss |
|---|---|---|---|---|---|---|---|---|---|---|---|
| 1 | 77.70 | 426.78 | 229.77 | 582.61 | 206.75 | 252.31 | 300.37 | 183.85 | 184.42 | 175.34 | 261.99 |
| 2 | 256.64 | 76.76 | 229.58 | 582.95 | 206.88 | 252.19 | 300.19 | 183.89 | 184.38 | 175.47 | 244.89 |
| 3 | 256.57 | 426.46 | 86.85 | 582.97 | 206.77 | 252.34 | 300.20 | 183.90 | 184.24 | 175.44 | 265.57 |
| 4 | 256.47 | 426.55 | 229.78 | 87.29 | 206.84 | 252.30 | 300.25 | 184.02 | 184.44 | 175.42 | 230.34 |
| 5 | 256.76 | 426.27 | 229.83 | 582.84 | 68.08 | 252.19 | 300.53 | 184.05 | 184.56 | 175.52 | 266.06 |
| 6 | 256.65 | 426.59 | 229.90 | 582.76 | 206.93 | 111.47 | 300.55 | 184.04 | 184.56 | 175.59 | 265.90 |
| 7 | 256.32 | 426.19 | 229.88 | 583.29 | 206.79 | 252.33 | 79.23 | 183.94 | 184.29 | 175.65 | 257.79 |
| 8 | 256.34 | 426.81 | 229.73 | 582.06 | 206.82 | 252.37 | 300.31 | 75.45 | 184.32 | 175.37 | 268.96 |
| 9 | 256.67 | 426.13 | 229.68 | 583.59 | 206.82 | 252.41 | 300.04 | 183.98 | 74.28 | 175.42 | 268.90 |
| 10 | 256.48 | 426.91 | 229.67 | 582.41 | 206.78 | 252.36 | 300.38 | 183.89 | 184.32 | 84.22 | 270.74 |
| multi-10 | 51.96 | 62.57 | 55.18 | 61.30 | 51.01 | 79.30 | 57.34 | 48.80 | 46.58 | 53.25 | 56.73 |

Table 8: The test MSE for the experiments from Section 4.3 for dSprites dataset.

| task | loss0 | loss1 | loss2 | loss3 | loss4 | loss5 | loss6 | loss7 | loss8 | loss9 | total_loss |
|---|---|---|---|---|---|---|---|---|---|---|---|
| 1 | 21.23 | 333.18 | 121.04 | 230.44 | 422.14 | 212.84 | 235.46 | 152.17 | 143.39 | 227.64 | 209.95 |
| 2 | 199.57 | 22.85 | 121.07 | 230.50 | 422.43 | 212.75 | 235.65 | 152.17 | 143.40 | 227.78 | 196.82 |
| 3 | 199.50 | 333.30 | 13.18 | 230.46 | 422.27 | 212.83 | 235.42 | 152.16 | 143.39 | 227.68 | 217.02 |
| 4 | 199.47 | 333.41 | 121.01 | 15.90 | 422.50 | 212.79 | 235.37 | 152.15 | 143.38 | 227.63 | 206.36 |
| 5 | 199.44 | 333.21 | 121.05 | 230.33 | 17.24 | 213.09 | 235.59 | 152.14 | 143.49 | 227.69 | 187.33 |
| 6 | 199.50 | 333.14 | 121.04 | 230.46 | 422.13 | 16.98 | 235.45 | 152.16 | 143.41 | 227.62 | 208.19 |
| 7 | 199.52 | 333.24 | 121.02 | 230.49 | 422.26 | 212.83 | 22.49 | 152.17 | 143.43 | 227.62 | 206.51 |
| 8 | 199.51 | 333.30 | 121.03 | 230.45 | 422.25 | 212.78 | 235.45 | 18.72 | 143.39 | 227.63 | 214.45 |
| 9 | 199.57 | 333.28 | 121.00 | 230.45 | 422.24 | 212.87 | 235.47 | 152.15 | 19.97 | 227.70 | 215.47 |
| 10 | 199.57 | 333.27 | 121.04 | 230.39 | 422.54 | 212.91 | 235.44 | 152.14 | 143.41 | 24.79 | 207.55 |
| multi-10 | 26.50 | 27.65 | 17.51 | 19.41 | 22.95 | 21.54 | 27.80 | 23.71 | 24.30 | 28.92 | 24.03 |

Table 9: The test MSE for the experiments from Section 4.3 for Shapes3D dataset.

| task | loss0 | loss1 | loss2 | loss3 | loss4 | loss5 | loss6 | loss7 | loss8 | loss9 | total_loss |
|---|---|---|---|---|---|---|---|---|---|---|---|
| 1 | 132.96 | 293.81 | 237.19 | 255.46 | 218.69 | 201.70 | 407.40 | 279.86 | 181.91 | 433.82 | 264.28 |
| 2 | 172.04 | 165.08 | 237.15 | 255.15 | 218.70 | 201.74 | 407.28 | 279.80 | 181.91 | 433.36 | 255.22 |
| 3 | 171.97 | 293.79 | 130.41 | 255.34 | 218.68 | 201.74 | 407.48 | 279.96 | 181.91 | 433.63 | 257.49 |
| 4 | 172.04 | 293.48 | 237.21 | 133.63 | 218.73 | 201.80 | 407.47 | 279.93 | 181.89 | 433.71 | 255.99 |
| 5 | 172.03 | 293.67 | 237.29 | 255.34 | 158.39 | 201.82 | 407.38 | 279.89 | 181.90 | 433.51 | 262.12 |
| 6 | 171.94 | 293.77 | 237.15 | 255.31 | 218.83 | 117.03 | 407.56 | 279.66 | 181.91 | 433.91 | 259.71 |
| 7 | 171.98 | 293.75 | 237.25 | 255.36 | 218.76 | 201.64 | 149.68 | 279.85 | 181.93 | 433.50 | 242.37 |
| 8 | 171.97 | 293.77 | 237.18 | 255.44 | 218.69 | 201.79 | 407.22 | 145.36 | 181.90 | 433.69 | 254.70 |
| 9 | 171.98 | 293.72 | 237.18 | 255.32 | 218.69 | 201.73 | 407.43 | 279.94 | 156.73 | 433.62 | 265.64 |
| 10 | 171.93 | 293.76 | 237.23 | 255.27 | 218.69 | 201.72 | 407.49 | 279.92 | 181.92 | 131.88 | 237.98 |
| multi-10 | 77.60 | 102.41 | 76.46 | 82.99 | 93.97 | 71.64 | 91.93 | 87.51 | 89.75 | 80.52 | 85.48 |

Table 10: The test MSE for the experiments from Section 4.3 for MPI3D dataset.

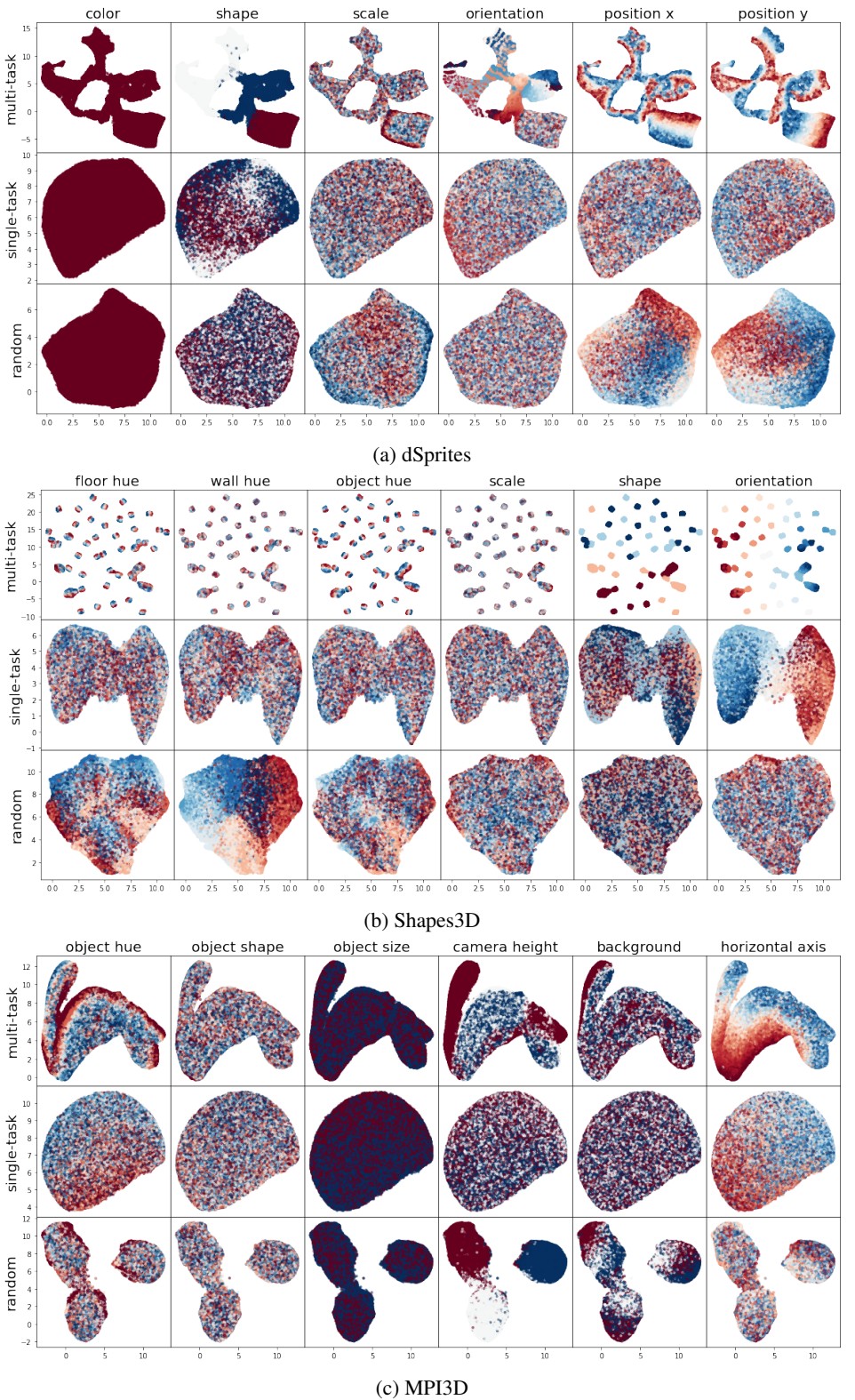

Figure 8: The UMAP embeddings obtained for untrained, single-task, and multi-task models on different datasets (computed on the test splits). The change in color corresponds to a change in the value of a selected true factor.

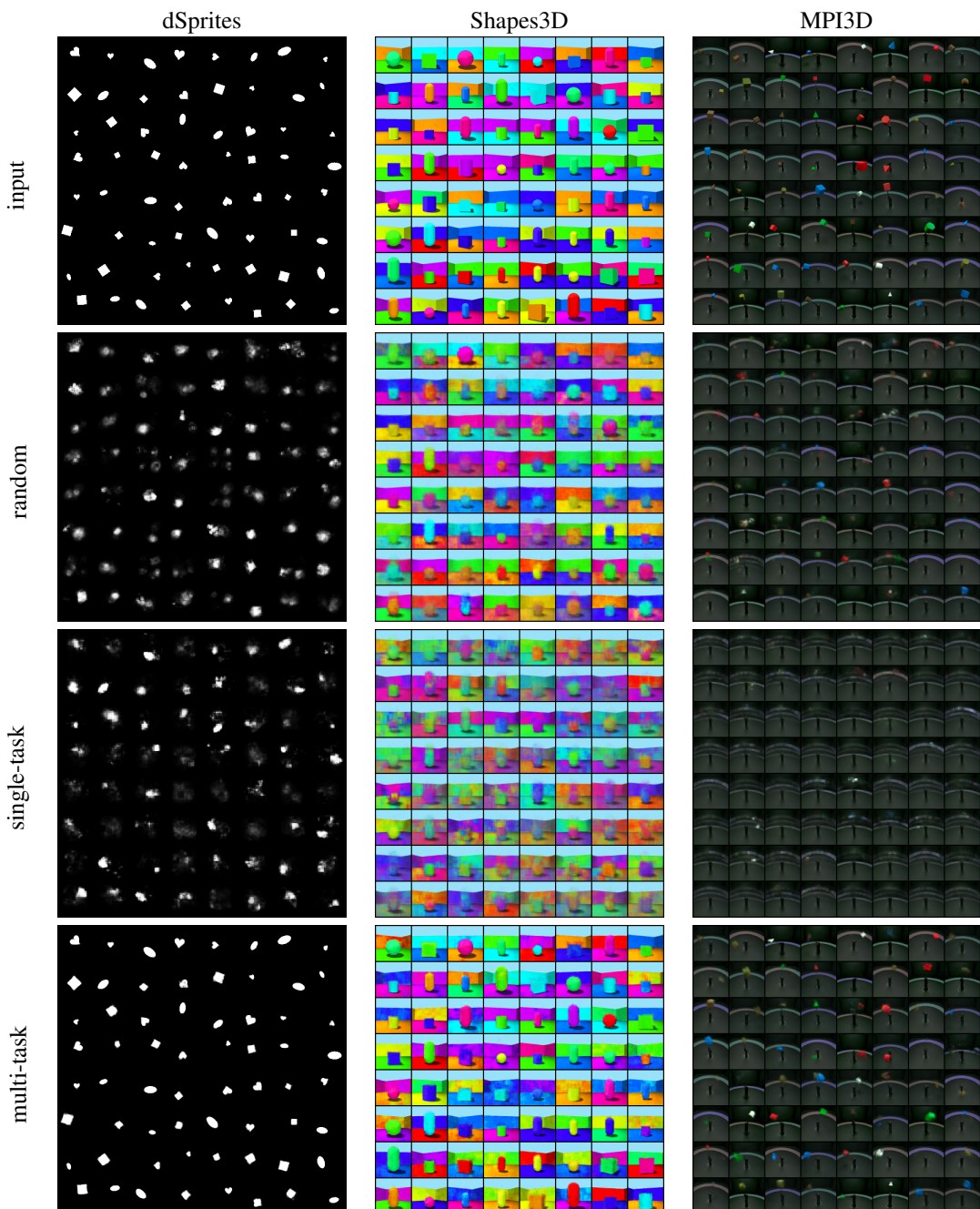

Figure 9: Reconstructions obtained during the experiments described in Section 4.2. The quality of the reconstruction for all datasets behaves similarly. One may easily observe that the multi-task encoder provided a latent space that can be successfully decoded into images that closely resemble the corresponding examples from the input. This is not the case in single-task or random encoders.

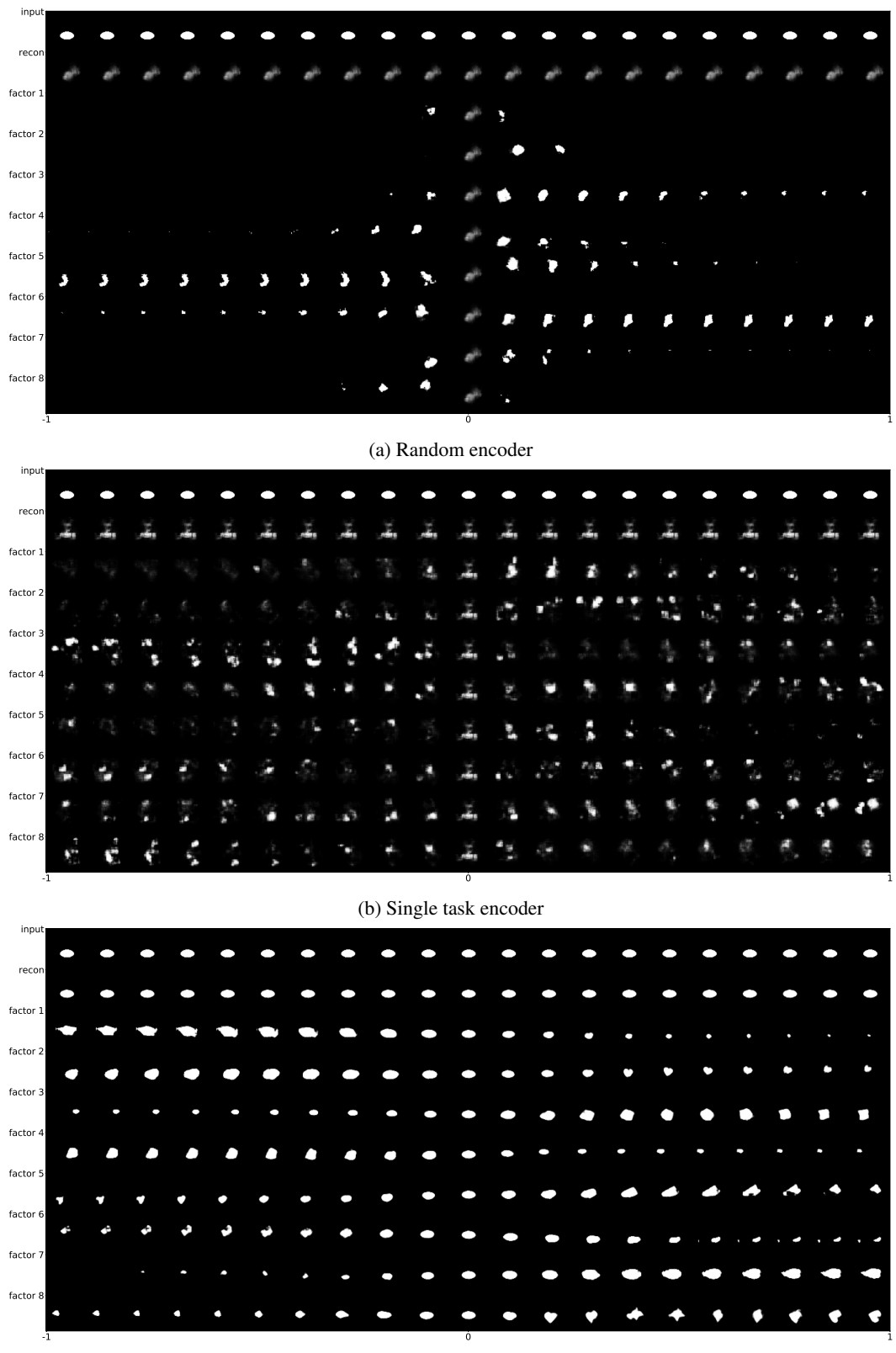

(a) Random encoder

(b) Single task encoder

(c) Multi-task encoder

Figure 10: Traverses for dSprites dataset over latent variable produced for a given architecture.

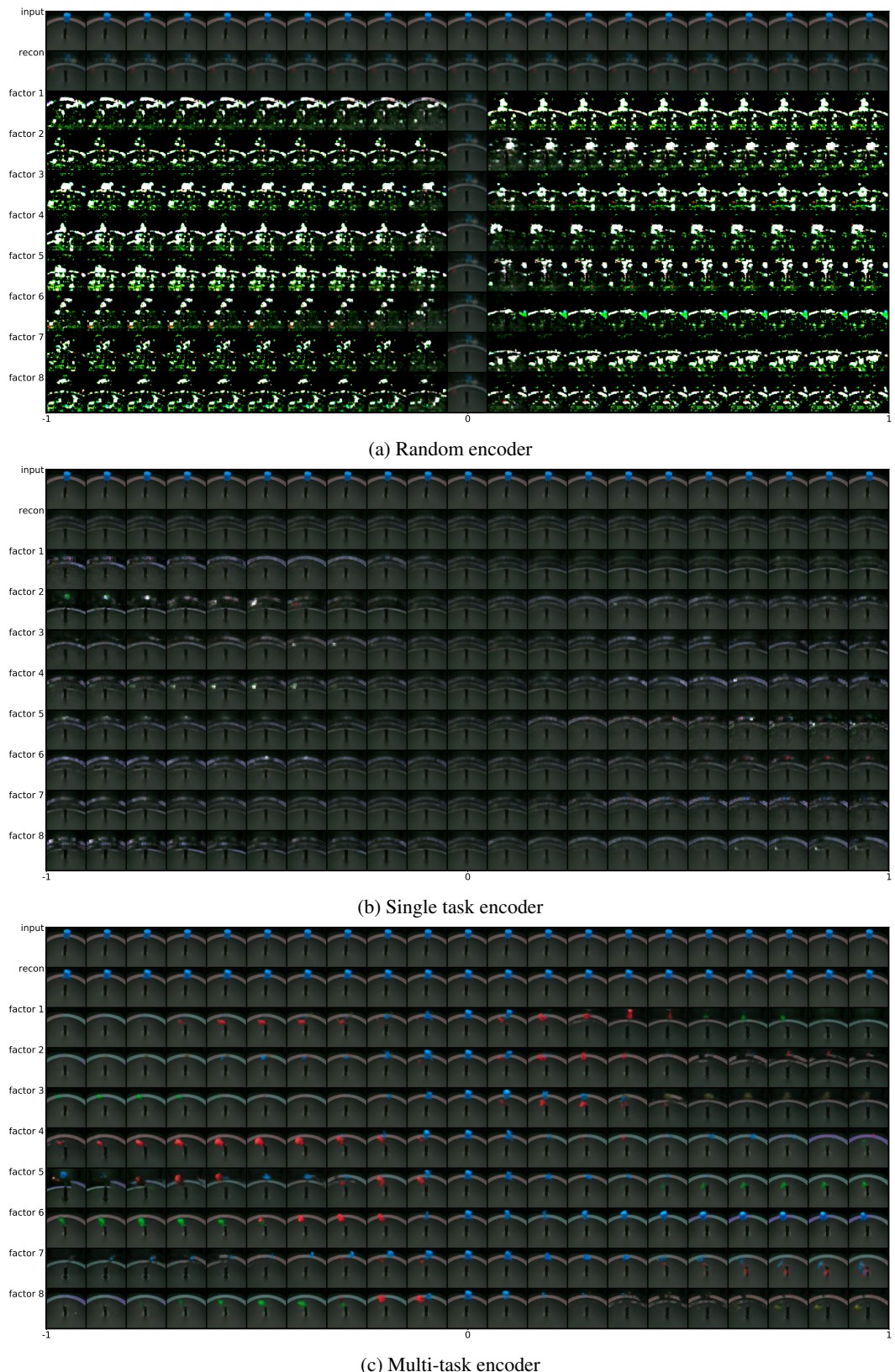

(a) Random encoder

(b) Single task encoder

(c) Multi-task encoder

Figure 11: Traverses for MPI3D dataset over latent variable produced for a given architecture.

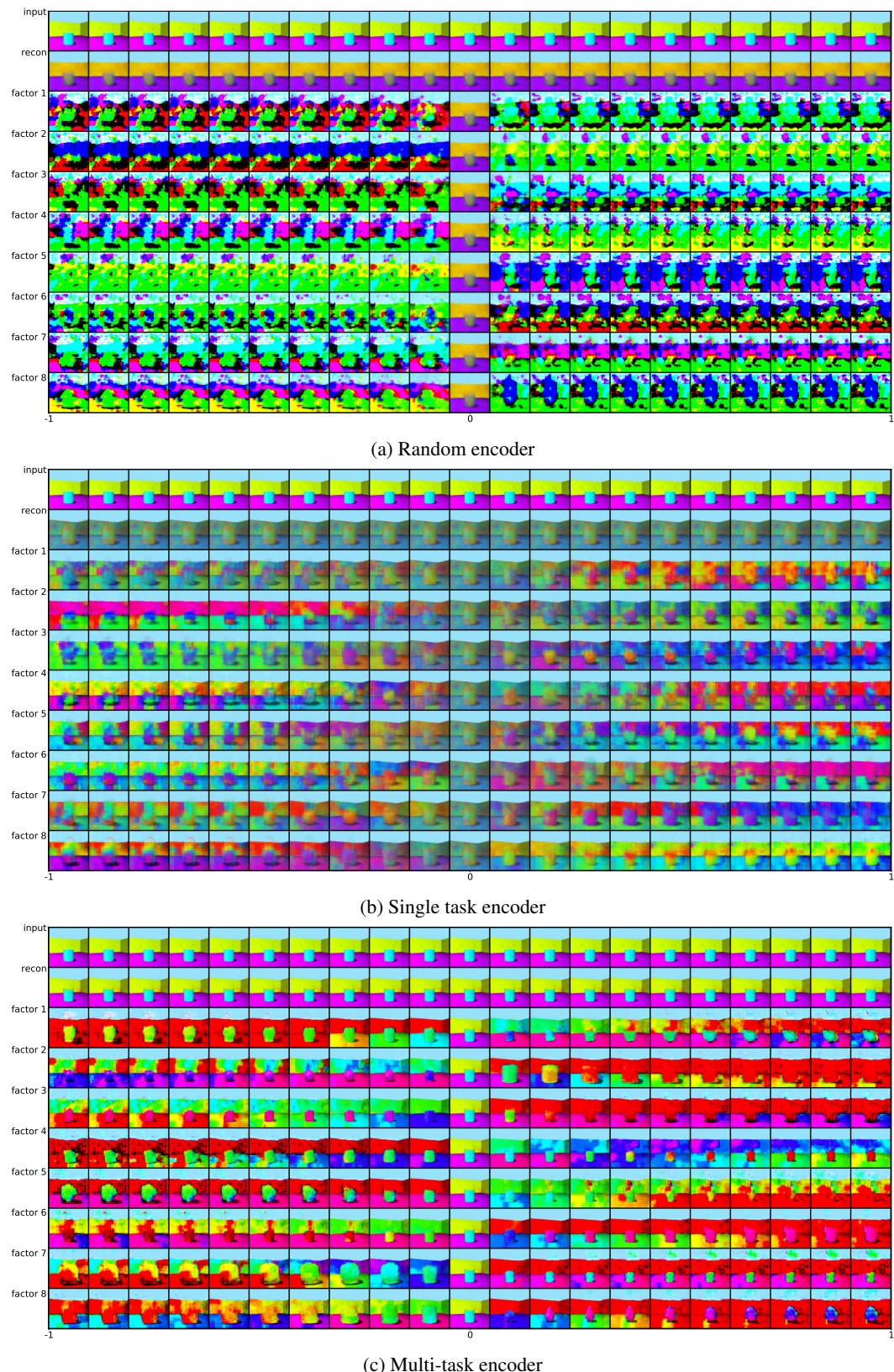

(a) Random encoder

(b) Single task encoder

(c) Multi-task encoder

Figure 12: Traverses for Shapes3D datset over latent variable produced for a given architecture.

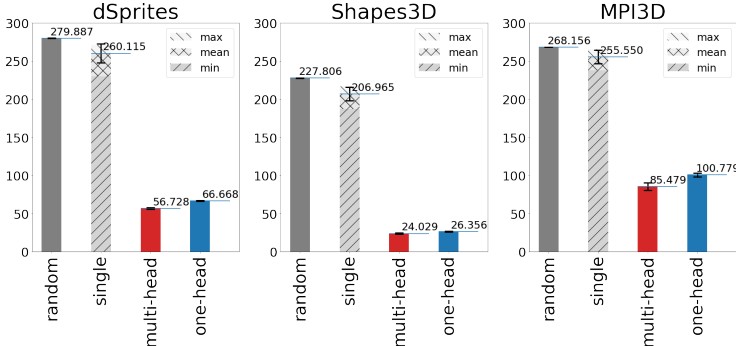

Figure 13: The average MSE on the test set computed for the random (untrained) model, a single-task model, and the multi-task models: multi-head and one-head. In the single-task case, we report the mean over all models for each task. The lower the value the better. As expected, the methods which jointly optimize the tasks achieve the best results.

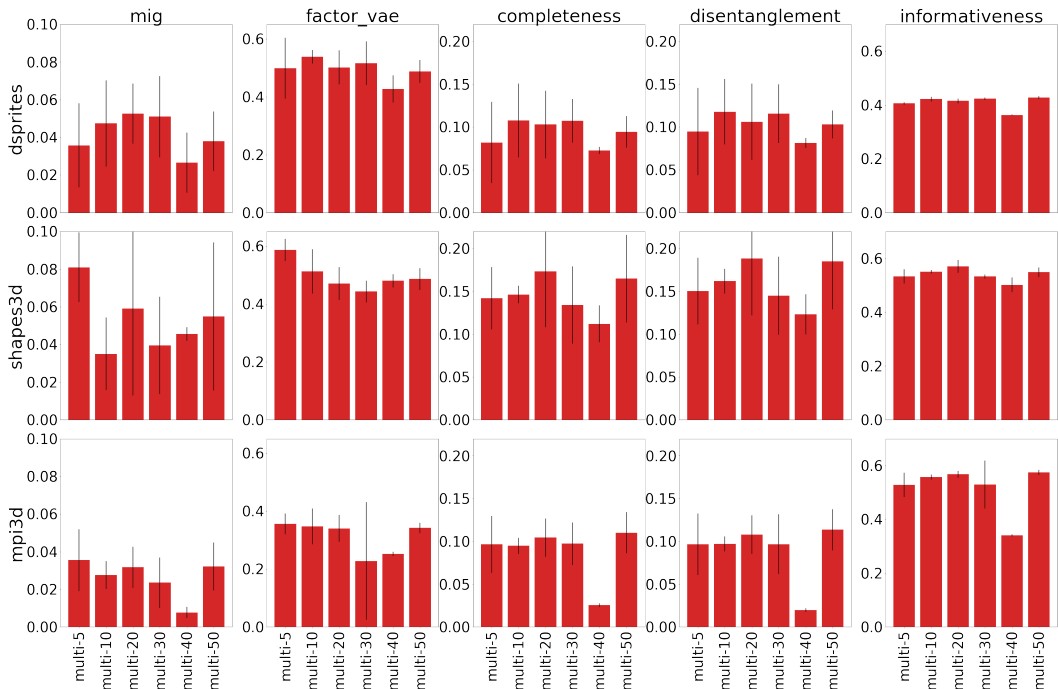

Figure 14: The disentanglement metrics computed for the multi-task model for different number of tasks presented on the x-axis. Experiments for the mpi3d dataset with 40 tasks did not converge (thus we observe a significantly lower values for this bar).

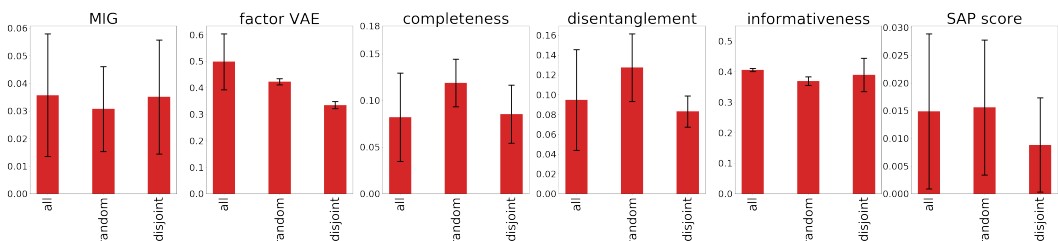

Figure 15: The disentanglement metrics on different factors splits in the multi-task seeting

