# OpenReview forum: "On the relationship between disentanglement and multi-task learning"
_ICLR.cc/2022/Conference — ICLR 2022 Submitted_

### Official Review · Reviewer_ZyfG · 2021-11-01

**Correctness:** 2
**Technical Novelty And Significance:** 3
**Empirical Novelty And Significance:** 2
**Recommendation:** 5
**Confidence:** 4

**Main Review:**

**Strengths:**
1. The paper is well-written and easy to read.

2. Authors attempted at analyzing how MTL affects the disentanglement quality, which is an interesting fundamental question that can be helpful to the community.

**Weaknesses:**
1. The most important issue with this paper is the experimental setup. It is absolutely clear that the amount of information about the generative factors encoded in the latent embedding will correlate with all of the disentanglement metrics used in this paper, regardless of whether the embedding is actually disentangled (disentanglement means that each factor is encoded by a subset of dimensions, and each dimension in the latent embedding depends only on one factor). All of the reported metrics compute an average across all generative factors, therefore, the more information is actually embedded into the latent representation, the higher are all of the disentanglement metrics. And, it is very clear, and even evident from the results illustrated in Figure 6, the MTL model encodes substantially more information about the input images than the single-task models. Therefore, the disentanglement results reported in Figure 3 are misleading, as it is impossible to compare the average disentanglement quality of the embeddings with various bandwidths. Even the illustration of embeddings in Figure 5 shows that the embeddings obtained by the multi-head model are still entangled, but it just has a different distribution.  A more conclusive metric should compute *the average number of factors* disentangled.

2. Another issue with the experimental setup is the experiment with single-head vs multi-head multi-task learning. First of all, it is not discussed that the single-head model might perform the tasks significantly worse, as it has much fewer parameters than the multi-head model. Basically, the single-head model will learn a representation that gives a good task-average performance, which will also lead to the loss of some amount of information. Additionally, since the task separation happens in the last layer of the head in the single-head case, the representation obtained by the layer before the last one might be more disentangled than the one before the head.

3.  The experimental setup with VAE-based embeddings for MTL is also inconclusive. The VAE-based multitask prediction quality should be compared with the vanilla hard sharing MTL model with a shared backbone that is trained from scratch on the input images. It is not clear how to interpret the results in Table 1, as it only implicitly tells us about the disentanglement quality of various VAEs, which seems out of the scope of this paper.



**Summary Of The Paper:**

The paper provides an empirical analysis of the effect of multi-task learning on the disentanglement of the learned latent representation. Authors use the existing synthetic datasets for the evaluation of disentanglement and create a set of tasks by predicting label values using randomly initialized MLPs of a fixed architecture. Authors report the experimental results on three synthetic datasets, comparing a hard parameter sharing strategy with a shared backbone and separate task-specific heads with the single-task case and randomly initialized networks. Additionally, the paper includes experimental results of MTL using the latent representations obtained by the three common disentanglement methods.

**Summary Of The Review:**

The paper aims to show that MTL implicitly improves the disentanglement quality of the learned representations, however, the crucial flaws in the experimental setup make all of the major results presented in this paper inconclusive.

---

> ### Author Response · Authors · 2021-11-18
> **(part1) Response to Reviewer ZyfG**
>
> We would like to thank the reviewer for the review and valuable feedback.
>
> > The most important issue with this paper is the experimental setup. It is absolutely clear that the amount of information about the generative factors encoded in the latent embedding will correlate with all of the disentanglement metrics used in this paper, regardless of whether the embedding is actually disentangled (disentanglement means that each factor is encoded by a subset of dimensions, and each dimension in the latent embedding depends only on one factor). All of the reported metrics compute an average across all generative factors, therefore, the more information is actually embedded into the latent representation, the higher are all of the disentanglement metrics. And, it is very clear, and even evident from the results illustrated in Figure 6, the MTL model encodes substantially more information about the input images than the single-task models. Therefore, the disentanglement results reported in Figure 3 are misleading, as it is impossible to compare the average disentanglement quality of the embeddings with various bandwidths. Even the illustration of embeddings in Figure 5 shows that the embeddings obtained by the multi-head model are still entangled, but it just has a different distribution. A more conclusive metric should compute the average number of factors disentangled.
>
> The reviewer points out that the metrics used in the setup are correlated with the informativeness of the representation. However, uncovering only independent factors (i.e. considering “disentanglement” alone) is an ill-posed problem, as there exist infinitely many solutions (Hyvärinen 1999, Locatello 2019b). Requiring the representation to be informative about the input plays the role of a restriction criterion that, hopefully, reduces the search space. Therefore, as disentanglement without informativeness makes no sense, we view informativeness as an important part of the uncovered representation. The used metrics also take into account different required properties of the representations, so for instance, a representation uncovering only one factor of variation and giving uncorrelated noise on the other components would have very low informativeness but obtain high independence-related metrics. We, therefore, understand that the concern of the reviewer lies in the situation in which the disentanglement is not present either in the multi-task nor single-task setup.
>
> We present an approach to calculating the number of disentangled factors in the appendix (section I). For each of the components of the studied representation, we compute the Spearman correlation with every ground truth factor and select the component for which the correlation is the largest and also statistically significant. Next, we return the number of unique factors matched in this way.  The results show that multi-task representations are able to recover more factors. In general, we agree that the interplay between informativeness and disentanglement is a very important issue and it is something that we will better highlight in our paper.
>
> References:
>
> Locatello, Francesco, et al. "Challenging common assumptions in the unsupervised learning of disentangled representations." international conference on machine learning. PMLR, 2019b
>
> Hyvärinen, Aapo, and Petteri Pajunen. "Nonlinear independent component analysis: Existence and uniqueness results." Neural networks 12.3 (1999): 429-439.
>
> > Another issue with the experimental setup is the experiment with single-head vs multi-head multi-task learning. First of all, it is not discussed that the single-head model might perform the tasks significantly worse, as it has much fewer parameters than the multi-head model. Basically, the single-head model will learn a representation that gives a good task-average performance, which will also lead to the loss of some amount of information. Additionally, since the task separation happens in the last layer of the head in the single-head case, the representation obtained by the layer before the last one might be more disentangled than the one before the head.
>
> The point of this experiment was different. We wanted to compare different techniques of multi-task problem solving to prove that in any of those strategies single-task is outperformed in the case of achieved disentanglement. We do not argue that for both of those strategies disentangled representations cannot be hidden on other layers. Our goal was to compare “pre-head” representations of multi-task models with the same layer of single-task models.

---

> > ### Author Response · Authors · 2021-11-18
> > **(part2) Response to Reviewer ZyfG**
> >
> > > The experimental setup with VAE-based embeddings for MTL is also inconclusive. The VAE-based multitask prediction quality should be compared with the vanilla hard sharing MTL model with a shared backbone that is trained from scratch on the input images. It is not clear how to interpret the results in Table 1, as it only implicitly tells us about the disentanglement quality of various VAEs, which seems out of the scope of this paper.
> >
> > In section 4.3 we discuss the impact of disentangled representations provided to the MTL artificially. Comparing the performance obtained by using ground truth factors versus a representation from a disentanglement model (which most likely would never achieve pure disentanglement) still provides us information about how the quality of the representation (in terms of uncovered factors) influences the training. Nevertheless, we provide the test losses of the fully trained MTL models in the appendix in Figure 13. and we can move those results also to Table 1 to provide a baseline comparison. In general, one can observe that having completely disentangled representation (i.e. the ground truth factors) is not necessarily indicative of a better performance.

---

> > > ### Author Response · Authors · 2021-11-29
> > > **Is there anything else the reviewer wants us to address?**
> > >
> > > Since the rebuttal period is closing soon, we would like to ask the Reviewer if our response addressed all of the points of the review. Also, please let us know if you have any further questions, as we would be happy to discuss them. Our main goal is the quality of the paper and your comments would help us improve it further.

---

> > > > ### Comment · Reviewer_ZyfG · 2021-12-03
> > > > **the paper still needs some improvement**
> > > >
> > > > I would like to thank the authors for a detailed response.
> > > >
> > > > >>We present an approach to calculating the number of disentangled factors in the appendix (section I). For each of the components of the studied representation, we compute the Spearman correlation with every ground truth factor and select the component for which the correlation is the largest and also statistically significant. Next, we return the number of unique factors matched in this way. The results show that multi-task representations are able to recover more factors.
> > > >
> > > > These results are also important for the analysis and should appear in the main paper. Overall, I share the concerns with the Reviewer r7PG that the idea of showing that MTL leads to more information being kept in the representations, not necessarily to disentangled representations. These two components must be separated in the analysis, and the paper in its current form does not seem to make this distinction.
> > > >
> > > > >> In general, one can observe that having completely disentangled representation (i.e. the ground truth factors) is not necessarily indicative of a better performance.
> > > >
> > > > These conclusions cannot be drawn from the results presented in section 4.3. It is not clear how the disentanglement correlates with the RMSE, because the disentanglement quality is not reported.
> > > >
> > > > Overall, I think the overall idea behind this paper is promising and would provide important insight into the MTL and representation learning fields, but in its current state, the paper does not present strong enough evidence to support the main claims. I also agree with the reviewers that the experiments with the more realistic tasks must be conducted.
> > > >
> > > > Therefore, I change my score to 5, leaning towards rejecting the paper.

---

### Official Review · Reviewer_r7PG · 2021-11-02

**Correctness:** 3
**Technical Novelty And Significance:** 3
**Empirical Novelty And Significance:** 4
**Recommendation:** 6
**Confidence:** 3

**Main Review:**

Strengths:
- The paper presents a reasonable evaluation setup (e.g. they include lots of different metrics, which I liked)
- They introduce a new dataset that might be interesting to the research community. (It's perhaps a little simplistic, but it seems like a fine starting point.)
- They show some fairly interesting findings -- most importantly, that multi-task learning seems to result in more disentangled representations (across every quantitative metric, and apparently wrt qualitative evaluations as well), but that the reverse isn't true (disentanglement doesn't clearly help with learning good multi-task representations) -- which I found fairly compelling.

Weaknesses:
- The paper makes some claims about the relationship between disentanglement and multi-task learning that I found somewhat uncompelling, or at least confusing. For example, they say that "Intuitively, a disentangled representation encompasses all the factors of variation and as such can be used for various tasks based on the same input space". However, I don't think of disentanglement as necessarily recovering *all* of the factors of variation. For example, presumably beta-VAE only disentangles a small fraction of the factors of variation in natural images, not all of them. So I think it's intuitive that multi-task models should "discard" less than single-task models, but I don't see how that intuitively connects to disentanglement. I suggest clarifying these sorts of claims.
- I thought the traversals experiment (figure 7) was kind of weak. I think the claim is very plausible, but I think it's hard to interpret this comparison because the reconstructions for the random and single-task encoders are much messier than for the multi-task encoder, and because it's just a single qualitative example (that's plausibly cherry picked).
- I would have liked some more realistic, large-scale, practical experiments in addition to the synthetic experiments they included in the paper (e.g. can we actually improve the disentanglement of large image generation models by doing multi-task training? What can we say about the features of "naturally" multi-task models, such as large pre-trained language models?)

Overall I think this paper has a lot of room for improvement, but is already interesting and reasonably thorough, so overall I would lean slightly towards acceptance.

**Summary Of The Paper:**

This paper studies the connection between disentanglement and multi-task learning. They present several sources of evidence that multi-task learning makes features quite a bit more disentangled. They also show some additional findings, such as that the reverse is not clearly true: make features more disentangled does not robustly help with multi-task learning.

**Summary Of The Review:**

Overall the main claim of the paper is interesting and moderately well supported, so while the paper clearly has room for improvement, I would still lean slightly towards acceptance.

---

> ### Author Response · Authors · 2021-11-18
> **(part1) Response to Reviewer r7PG**
>
> We would like to thank the reviewer for the review and valuable comments.
>
> > The paper makes some claims about the relationship between disentanglement and multi-task learning that I found somewhat uncompelling, or at least confusing. For example, they say that "Intuitively, a disentangled representation encompasses all the factors of variation and as such can be used for various tasks based on the same input space". However, I don't think of disentanglement as necessarily recovering all of the factors of variation. For example, presumably beta-VAE only disentangles a small fraction of the factors of variation in natural images, not all of them. So I think it's intuitive that multi-task models should "discard" less than single-task models, but I don't see how that intuitively connects to disentanglement. I suggest clarifying these sorts of claims.
>
> We use the definition of disentanglement from (Bengio, 2013) which states that each component of a disentangled representation corresponds to a different factor of variation in the data. Imagine, for example, two tasks on a shapes3d-like dataset, one depending on the position and shape of the object, and the other one depending on the position and the color. Then when performing multi-task learning one could intuitively presume that it would be more beneficial to give the model the disentangled factors (position, shape, color) rather than a pixel image. In other words, the disentangled representation would contain all the factors of variations needed for the model. However, we agree that in practice obtaining disentanglement is very difficult or even impossible (Locatello, 2019), and models such as beta-VAE are only able to retrieve a fraction of factors (or independent, entangled factors) which we would consider a “partially disentangled” representation. At the same time, we think that a representation retrieving a significant number of factors of variation (even though not all of them) should be intuitively useful for multi-task learning since there is a high chance that it will contain the factors of variation required for a particular task.
>
> > I thought the traversals experiment (figure 7) was kind of weak. I think the claim is very plausible, but I think it's hard to interpret this comparison because the reconstructions for the random and single-task encoders are much messier than for the multi-task encoder, and because it's just a single qualitative example (that's plausibly cherry picked).
>
> Traversal visualisations are standard qualitative examples in disentanglement literature (Locatello et al., 2019c; Kumar et al., 2017; Sanchez et al., 2019; Sorrenson et al., 2020; Locatello et al., 2019b). We took this approach to visualise the latent reconstructed from different models. We do not argue that those representations retrieved by our single or multi-task models are any better or even comparable with the representations recovered by SOTA models in this research field. The purpose of this experiment was just to visually assess the quality of the obtained representations between the multi-task and single task setups. The ability of reconstruction tells us about the informativeness of the representations (whether we are able to retrieve the input), while the traversals should give intuition on what factors of variation are present in each component. The strongest point of our claim that representations found by multi-task models are more disentangled (and subjectively meaningful) than their single-task counterpart is that this result can be extrapolated on the other datasets (see fig. 10, 11, 12 from the appendix). Note that this section is not a stand-alone experiment, but rather an additional discussion of the quantitative results presented in the study made in section 4.1.
>
>
> We would also like to point out that the results were not cherry-picked but randomly selected from the dataset (one can see more examples in the appendix for all used datasets). We could provide more examples to assure that any of those were not cherry-picked. Additionally, we provided code for the reproduction of the experiments in additional materials attached to this paper.

---

> > ### Author Response · Authors · 2021-11-18
> > **(part2) Response to Reviewer r7PG**
> >
> > > I would have liked some more realistic, large-scale, practical experiments in addition to the synthetic experiments they included in the paper (e.g. can we actually improve the disentanglement of large image generation models by doing multi-task training? What can we say about the features of "naturally" multi-task models, such as large pre-trained language models?)
> >
> > Note that in order to investigate the relationship between multi-task learning and disentanglement, we require a dataset that gives access to true generating factors z (apart from the observations x) and proposes multiple tasks depending on z. The first condition is required in order to make sure how well the learned representations approximate the true latent factors z (having the access to true factors allows us to use supervised disentanglement metrics).  The second condition is needed to train a network on multiple non-trivial tasks. As described in section 3.1, we are not aware of any non-trivial datasets that would fulfill both of those conditions, therefore we decide to perform the analysis on synthetic tasks. Note that one could try to drop the first condition by trying to use unsupervised metrics for disentanglement learning, but as discussed in section 2.1, 3.3 (and reported, for instance in Locatello 2019b), unsupervised metrics often underperform or give contradictory results, preventing a fair analysis.
> > Furthermore, a benefit of a synthetic task is that we have full control over the dataset setup. This allows us to test our hypothesis more carefully. Note that despite the fact that the tasks are synthetic, we can still say something about the expected behaviour in a large-scale scenario. For instance, the reviewer asks whether disentanglement can be improved by doing multi-task training. By looking at the results from section 4.1 and comparing them to the values obtained by methods dedicated only for disentanglement (see Locatello 2019), it is clear that the multi-task representations, although better than the single-task ones, obtain much worse performance.
> > Finally, as pointed out by the reviewer, our study is indeed limited to the image datasets. This is due to the fact that most quintessential disentanglement datasets (such as dSprites or Shapes3D) are of this type. Defining disentanglement, for instance, in language-based models is much less trivial (what would be a generating factor of a sentence?). This is of course a very interesting topic, but so vast that we consider it a research area suitable for future work.
> >
> > References:
> >
> > Bengio, Yoshua. "Deep learning of representations: Looking forward." International conference on statistical language and speech processing. Springer, Berlin, Heidelberg, 2013
> >
> > Locatello, Francesco, et al. "Challenging common assumptions in the unsupervised learning of disentangled representations." international conference on machine learning. PMLR, 2019b
> >
> >
> > Locatello, Francesco, et al. "Disentangling factors of variation using few labels." arXiv preprint arXiv:1905.01258 2019c.
> >
> >
> > Kumar, Abhishek, Prasanna Sattigeri, and Avinash Balakrishnan. "Variational inference of disentangled latent concepts from unlabeled observations." arXiv preprint arXiv:1711.00848 (2017). ICLR 2018
> >
> >
> > Sanchez, Eduardo Hugo, Mathieu Serrurier, and Mathias Ortner. "Learning disentangled representations via mutual information estimation." European Conference on Computer Vision. Springer, Cham, 2020.
> >
> >
> > Sorrenson, Peter, Carsten Rother, and Ullrich Köthe. "Disentanglement by nonlinear ica with general incompressible-flow networks (gin)." arXiv preprint arXiv:2001.04872 2020.

---

> > > ### Author Response · Authors · 2021-11-29
> > > **Is there anything else the reviewer wants us to address?**
> > >
> > > Since the rebuttal period is closing soon, we would like to ask the Reviewer if our response addressed all of the points of the review. Also, please let us know if you have any further questions, as we would be happy to discuss them. Our main goal is the quality of the paper and your comments would help us improve it further.

---

> > > > ### Comment · Reviewer_r7PG · 2021-11-29
> > > > **Thanks**
> > > >
> > > > Thank you for your detailed response. I found your first explanation in response to my comment clearer than what was written in the paper. I think the point about traversals is ok, but I still have some concerns -- I don't have an objection to traversal experiments in general, I just wanted to point out that these particular traversal results weren't as compelling as I might have hoped. For the realistic experiments, I was imagining a qualitative evaluation setup (e.g. using traversals), which wouldn't necessarily require having access to the true generating factors. I hope these clarifications are helpful.
> > > >
> > > > Overall I'll keep my score of 6.

---

### Official Review · Reviewer_zzap · 2021-11-03

**Correctness:** 3
**Technical Novelty And Significance:** 3
**Empirical Novelty And Significance:** 3
**Recommendation:** 5
**Confidence:** 4

**Main Review:**

The work is quite interesting, and I think fundamental analyses of multitask learning representations like this are important within multitask learning research. The analysis goes through multiple different network architectures and tabulates different metrics, and is reasonably thorough. The visualizations are also clear and fairly convincing.

My concerns are mostly in the problem setup. Currently my rating will be a weak reject but I am happy to upgrade my score if authors can address some of these concerns satisfactorily.

(1) In the vast majority of multitask settings, the output tasks are correlated. However, since the output regression tasks here are generated through multiple IID randomly initialized networks, this seems to imply no correlation between these tasks. How would stronger correlation between multitask outputs affect the results? Happy to see either theoretical or empirical analysis on this. And why do the authors think the current results would apply to real applications of multitask learning, given this discrepancy?

(2) Why are the single-task network results so weak, even compared to a randomly initialized network? This seems like a weird result to me, since a random network should be just outputting noise, and yet the claim here is that the single-task network is actually performing worse in terms of the disentanglement metrics. Could authors provide loss curves for the multitask regression tasks and show that the single-task networks are at least training? I feel like the weird random vs single results might suggest the single-task network is not well tuned or the synthetic tasks are too noisy.

(3) As an addendum to the above, often for multitask networks the single-task networks perform on-par or even slightly better than the multitask network on pure loss metrics. Can authors provide an explanation as to why there's such a huge discrepancy here?

(4) Why did authors choose not to compare to the approach of regressing the latent variables z directly? I would expect such a network would perform quite well. In a related question, is it plausible that the improved performance with multiple tasks is more a function of noise reduction given the extra data? Have authors tried to dramatically reduce the size of the generating functions for the synthetic labels and see if the conclusions still hold?

(5) Why are the randomly initialized networks still producing reasonable reconstruction results? I was under the impression that the encoder is a straightforward convnet with no skip connections but am I wrong on that?

(6) Authors should greatly expand their related work section. There is a rich abundance of research out there in regards to multitask learning within a deep learning context and authors only touched the surface. In particular, how do the current disentanglement ideas relate to recent work in improving deep multitask networks?

**Summary Of The Paper:**

Authors provide disentanglement analysis of multitask learning by creating a semi-synthetic dataset based on latent information in other simple datasets. Authors run the latent information through randomly initialized FC layers to create auxiliary tasks, and then train a convolutional network to produce a bottlenecked representation which is then fit to these auxiliary tasks. Authors also train various autoencoder models to see if disentangled representations help with multitask training, with results inconclusive.

**Summary Of The Review:**

The idea is interesting and I would love to see more papers try to tackle fundamental analysis of multitask models like this, but there are enough confusing and counterintuitive results that I would like to clarify things before endorsing this work. I very much look forward to the author's response.

---

> ### Author Response · Authors · 2021-11-18
> **(part1) Response to Reviewer zzap**
>
> Thank you for the review and valuable feedback.
>
> > (1) In the vast majority of multitask settings, the output tasks are correlated. However, since the output regression tasks here are generated through multiple IID randomly initialized networks, this seems to imply no correlation between these tasks. How would a stronger correlation between multitask outputs affect the results? Happy to see either theoretical or empirical analysis on this. And why do the authors think the current results would apply to real applications of multitask learning, given this discrepancy?
>
> We empirically analyzed the correlations present in our dataset and found out that the tasks are in fact correlated to a certain degree (for example, for the Shapes3D-based dataset, the highest correlation is around 0.5, and the average absolute correlation is approximately 0.15). We assume this is due to the fact that the functions we are generating belong to the same family (i.e. the architecture and the distribution we sample the weights from are constant), which makes it possible for certain pairs of tasks to become correlated. Moreover, all functions share some information since they are all using the same input. As such, we believe that our datasets exhibit enough correlations to approximate real-world multi-task settings, however, we would be happy to extend this analysis if so requested.
>
> > (2) Why are the single-task network results so weak, even compared to a randomly initialized network? This seems like a weird result to me, since a random network should be just outputting noise, and yet the claim here is that the single-task network is actually performing worse in terms of the disentanglement metrics. Could authors provide loss curves for the multitask regression tasks and show that the single-task networks are at least training? I feel like the weird random vs single results might suggest the single-task network is not well-tuned or the synthetic tasks are too noisy.
>
> > (3) As an addendum to the above, often for multitask networks the single-task networks perform on-par or even slightly better than the multitask network on pure loss metrics. Can authors provide an explanation as to why there's such a huge discrepancy here?
>
> Ad (2) and (3): Note that the results presented in Fig. 3 and Fig. 13 for the single tasks are the mean results across all tasks. Therefore, for example in Fig. 13, the total average loss across all tasks can be high, since the single task models optimize only for a single task and have worse performance on all the other tasks (close to that of a randomly initialized network). Of course, the performance of the single-task network on the task on which it was trained is much better (we have added all the results in the appendix in section J).

---

> > ### Author Response · Authors · 2021-11-18
> > **(part2) Response to Reviewer zzap**
> >
> > > (4) Why did authors choose not to compare to the approach of regressing the latent variables z directly? I would expect such a network would perform quite well. In a related question, is it plausible that the improved performance with multiple tasks is more a function of noise reduction given the extra data? Have authors tried to dramatically reduce the size of the generating functions for the synthetic labels and see if the conclusions still hold?
> >
> > Ad (4): To this end as, as highlighted in section 3.1, we need a dataset that has access both to the target values and true latent components. Setting the targets to the true latent components would be equal to explicitly training a supervised disentanglement model, and would tell us nothing about the possibility of implicit disentanglement, as a consequence of the inductive bias of multi-task learning.  In other words, our aim was not to perform a supervised disentanglement, but rather to measure whether disentanglement appears in the representations used by a multi-task model.
> >
> > However, we did check whether having a completely disengaged representation (i.e. the ground truth factors) helps in multi-task learning (note that this is an inverse question to the one discussed above). Such an experiment is highlighted in section 4.3, where we analyze ground truth factors (the generating latent z for a given dataset) that serve as an input for a regression directly. Such an approach leads to better results than the use of the representations learned by disentanglement models.
> >
> > (is it plausible that the improved performance with multiple tasks is more a function of noise reduction given the extra data?) The amount of data used for training single and multi-task models were the same (experimental setup is fully described in appendix section C.1). Intuition behind multi-task learning is that inductive transfer improves generalization by using the domain information contained in the training signals of related tasks as an inductive bias. It does this by learning tasks in parallel while using a shared representation; what is learned for each task can help other tasks be learned better (Caruana, 1997). In this sense, the multi-task regression model is always a noise reduction one.
> >
> > (Have authors tried to dramatically reduce the size of the generating functions for the synthetic labels and see if the conclusions still hold?). We did not try to reduce the size of the generating functions. Note, that as discussed in “Why did authors choose not to compare to the approach of regressing the latent variables z directly”, we do not want the tasks to be too similar to the true factors, but rather to model functions of such factors. We did, however, perform an experiment in which we vary the number of factors taken as the input to the generating function. I.e. instead of using all factors as the input to the network, we also experiment with taking a random subset of the factors for each task and taking disjoint subsets for each odd and even task (see section H in the appendix).
> >
> > > (5) Why are the randomly initialized networks still producing reasonable reconstruction results? I was under the impression that the encoder is a straightforward convnet with no skip connections but am I wrong on that?
> >
> > In this experiment, the “random” initialization of the encoder was fixed throughout the training (In concept, this could be similar to projecting the inputs on a space given by a fixed random transformation). Such obtained representations were then passed to the decoder, which was trainable. Therefore the reconstructions produced by such an approach could still be quite informative about the input.
> >
> > > (6) Authors should greatly expand their related work section. There is a rich abundance of research out there in regards to multitask learning within a deep learning context and authors only touched the surface. In particular, how do the current disentanglement ideas relate to recent work in improving deep multitask networks?
> >
> > Due to the constraint on the length of the paper, we focused on the most important and relevant research subjects - in our opinion. Since the goal of the paper was to investigate what is the relationship between multi-task (via hard-parameter sharing) and disentanglement, and not to propose any new technique for learning multi-task models, we did not put so much emphasis on different deep learning approaches to this problem. However, we will gladly extend the related work section with a discussion on any papers the reviewer finds necessary.
> >
> > References:
> > Richard Caruana. Multitask learning: A knowledge-based source of inductive bias. In Proceedings of the Tenth International Conference on Machine Learning, pp. 41–48. Morgan Kaufmann, 1993.

---

> > > ### Author Response · Authors · 2021-11-29
> > > **Is there anything else the reviewer wants us to address?**
> > >
> > > Since the rebuttal period is closing soon, we would like to ask the Reviewer if our response addressed all of the points of the review. Also, please let us know if you have any further questions, as we would be happy to discuss them. Our main goal is the quality of the paper and your comments would help us improve it further.

---

> > > ### Comment · Reviewer_zzap · 2021-11-30
> > > **RE: Response**
> > >
> > > Thank you for the detailed response! I have carefully reviewed the response and re-read the paper and have decided to keep my rating the same. I think it is definitely an interesting work but some of the elements of the response have engendered additional questions - for example, it's very odd that correlations in the answer to (1) exist as even if the sampling is done from the same distribution the samplings themselves are still IID. I also do think loss curves and other diagnostics should be included and analyzed at least in supplementary. So I think there are still some technical details that create cause for concern here. I do hope that, should the authors not get their work through in this conference that they continue to work on it since I would love to see a more polished version of this published at a good venue.

---

### Official Review · Reviewer_NYHm · 2021-11-05

**Correctness:** 3
**Technical Novelty And Significance:** 2
**Empirical Novelty And Significance:** 3
**Recommendation:** 6
**Confidence:** 3

**Main Review:**

Major concerns:

I think the relationship between multi-task learning and disentanglement can be more carefully discussed. The number of learning tasks and also the relevance of different tasks could matter.

Throughout the paper, the authors use ten different random tasks. Did the authors try to vary the number of multi-tasks and see how that affects disentanglement? Consider a discrete generative factor (with m values); how does the number of different random tasks (n) help with disentanglement given n<m, n=m, and n>m?

Also, in the paper, there is no assumption on the relevance among different tasks. Given three discrete generative factors z_1, z_2 and z_3, if one task is classification with respect to z_1 and z_2, while the second task is the classification for z_2, the learned representation (probably) would not show clear disentanglement property on z_3. Also, in such kind of scenario, removing the second task probably would not hurt the quality of the learned representation in terms of disentanglement.

**Summary Of The Paper:**

This paper studies the relationship between disentanglement and multi-task learning (hard parameter sharing) via empirical study. The authors very carefully examined if multi-task learning encourages disentanglement. The authors performed an extensive empirical study and looked at different metrics on a couple of datasets.

What’s more, the authors also provided synthetic datasets that can help study the relationship between feature disentanglement and multi-task learning. The authors described the right approach for such kind of dataset synthesis.

**Summary Of The Review:**

After initial review, I am inclined to accept this paper; I am happy to update my score after discussing it with authors and other fellow reviewers.
------------------------------Post-rebuttal---------------------------------

I read the authors' responses and went through the authors' new experiments. I also read the reviews from other fellow reviewers and the authors' responses to them. At this moment, I am inclined to keep my score unchanged.

I think each task would encourage representations to respect specific transformations and thus may implicitly encourage disentanglement. Therefore, I believe the number of transformations and their relatedness matter; this is the motivation of my questions.

The authors try to address one of my questions on the number of tasks, but the conclusion is inconclusive. Regarding the experiments on exploring the effect of the number of tasks, it's unclear why 40 tasks is always a bad choice in all three datasets.

---

> ### Author Response · Authors · 2021-11-18
> **Response to Reviewer NYHm**
>
> We would like to thank the reviewer for the review and valuable comments.
>
> > Throughout the paper, the authors use ten different random tasks. Did the authors try to vary the number of multi-tasks and see how that affects disentanglement? Consider a discrete generative factor (with m values); how does the number of different random tasks (n) help with disentanglement given n<m, n=m, and n>m?
>
> We did perform experiments with different numbers of tasks. In particular, we tested n=5,10,20,30,40,50 for all the datasets. We have added these results to the appendix (see section G in the revised paper). However, it is difficult to make any clear conclusions about the effect of the number of tasks n on the metrics, as the increase of n (to some point) in some cases leads to higher values of selected metrics, at the same time having a negative impact on the others (see, for instance, Shapes3D  and factor_vae versus DCI disentanglement or DCI completeness). These discrepancies are also not consistent between datasets (consider the top row for the dSprites dataset versus the Shapes3D), although for all datasets disentanglement metrics for multi-task models are significantly better than the single-task ones.
>
> > Also, in the paper, there is no assumption on the relevance among different tasks. Given three discrete generative factors z_1, z_2 and z_3, if one task is classification with respect to z_1 and z_2, while the second task is the classification for z_2, the learned representation (probably) would not show clear disentanglement property on z_3. Also, in such kind of scenario, removing the second task probably would not hurt the quality of the learned representation in terms of disentanglement.
>
> We tested a couple of approaches with different distributions of generative factors and their impact on final disentanglement in a multi-task setting. In particular, apart from the approach presented in the paper which uses all of the factors to generate a task, we also considered a scenario in which a random subset of the factors is sampled for each task, and a scenario in which the tasks are generated from disjoint subsets (every odd task depends only on the first half of the factors and every even task on the other half). We compared these approaches and found out that in a multi-task setting the computed disentanglement measures vary and the precise subset of incorporated factors in the task generating procedure does not have any conclusive impact on the final quality of the learned representation. However, note that regardless of the number of used factors, in the majority of cases the results for multi-task are still better than in a single task approach (see Section H).

---

> > ### Author Response · Authors · 2021-11-29
> > **Is there anything else the reviewer wants us to address?**
> >
> > Since the rebuttal period is closing soon, we would like to ask the Reviewer if our response addressed all of the points of the review. Also, please let us know if you have any further questions, as we would be happy to discuss them. Our main goal is the quality of the paper and your comments would help us improve it further.

---

### Decision · Program_Chairs · 2022-01-20

**Decision:**

Reject

**Comment:**

This paper aims to look at the relationship between disentanglement
and multi-task learning.  The authors claim to show that disentanglement
emerges naturally from MTL.

The main discussion was whether the claim that disentanglement emerges
naturally from MTL has been adequately demonstrated.  The main
issue is that MTL results in more extraction of information and that
is hard to disentangle from the disentanglement metrics used.

Reviewers agreed the work was interesting but not as complete as would
be desirable.  I also feel it is not ready for ICLR presentation, but
with further work could be a nice future contribution.